# JAMBA: HYBRID TRANSFORMER-MAMBA LANGUAGE MODELS

**Jamba Team**

**AI21labs**

## ABSTRACT

We present Jamba, a novel hybrid Transformer-Mamba mixture-of-experts (MoE) architecture. Jamba interleaves blocks of Transformer and Mamba layers, enjoying the benefits of both model families. MoE is added in some of these layers to increase model capacity while keeping active parameter usage manageable. This flexible architecture allows resource- and objective-specific configurations. We implement two configurations: Jamba-1.5-Large, with 94B active parameters, and Jamba-1.5-Mini, with 12B active parameters. Built at large scale, Jamba models provide high throughput and small memory footprint compared to vanilla Transformers, especially at long-context tasks, with an effective context length of 256K tokens, the largest amongst open-weight models. At the same time, they are also competitive on standard language modeling and chatbot benchmarks. We study various architectural decisions, such as how to combine Transformer and Mamba layers, and how to mix experts, and show that some of them are crucial in large-scale modeling. To support cost-effective inference, we introduce ExpertsInt8, a novel quantization technique that allows fitting Jamba-1.5-Large on a machine with 8 80GB GPUs when processing 256K-token contexts without loss of quality. We also describe several interesting properties of this architecture that the training and evaluation of Jamba have revealed. The model weights are publicly available.

**Models:** https://huggingface.co/ai21labs

## 1 INTRODUCTION

We introduce Jamba, a novel hybrid architecture, which combines Transformer layers (Vaswani et al., 2017) with Mamba layers (Gu & Dao, 2023), a recent state-space model (Gu et al., 2021b;a), as well as a mixture-of-experts (MoE) module (Shazeer et al., 2017; Fedus et al., 2022). Jamba thus combines two orthogonal architectural designs that together give it improved performance and higher throughput, while maintaining a manageable memory footprint. We release two models based on this architecture: Jamba-1.5-Mini (12B active parameters, 52B total available parameters) and Jamba-1.5-Large (94B active, 398B total available parameters), which are respectively designed to fit on a single 80GB GPU and a single 8x80GB GPU machine. However, the Jamba architecture supports other design choices, depending on one's hardware and performance requirements.

The fundamental novelty of Jamba is its hybrid Transformer-Mamba architecture. Despite the immense popularity of the Transformer as the predominant architecture for language models, it suffers from two main drawbacks. First, its high memory and compute requirements hinder the processing of long contexts, where the key-value (KV) cache size becomes a limiting factor. Second, its lack of a single summary state entails slow inference and low throughput, as each generated token performs a computation on the entire context. In contrast, older recurrent neural networks (RNN) (Elman, 1990; Hochreiter & Schmidhuber, 1997; Mikolov et al., 2010), which summarize an arbitrarily long context in a single state, do not suffer from these limitations. RNN models have their own shortcomings, however. They are costly to train as training cannot be parallelized across time. And they struggle with long distance relationships, which the hidden state captures to a limited extent.

Recent state space models (SSMs) like Mamba are more efficient to train than RNNs and are more capable at handling long distance relationships, but still lag behind the performance of comparably sized Transformer language models. Taking advantage of both model families, Jamba combines Transformer and Mamba layers, at a certain ratio. Varying the ratio of Transformer/Mamba layers allows balancing memory usage, efficient training, and long context capabilities.

A few other recent attempts to combine Attention and SSM modules are worth noting. Zuo et al. (2022) mixes an S4 layer (Gu et al., 2021a) with a local attention layer, followed by a sequence of local attention layers; it shows experiments with small models and simple tasks. Gu & Dao (2023) report that interleaving Mamba and attention layers is only slightly better than pure Mamba in terms of perplexity, with models up to 1.3B parameters. Pilault et al. (2023) start with an SSM layer followed by chunk-based Transformers, with models up to 1.3B showing improved perplexity. Fathullah et al. (2023) adds an SSM layer before the self-attention in a Transformer layer, while Saon et al. (2023) add the SSM after the self-attention, both showing improvements on speech recognition. Park et al. (2024) replace the MLP layers in the Transformer by Mamba layers, showing benefits in simple tasks. These efforts are different from Jamba both in the particular way in which the SSM component is mixed with the attention one, and in the scale of implementation. Closest are perhaps H3 (Fu et al., 2022), a specially designed SSM that enables induction capabilities, and a generalization called Hyena (Poli et al., 2023a). The former proposed a hybrid architecture that replaces the second and middle layers with self-attention, and was implemented with up to 2.7B parameters and 400B training tokens. However, its perfomance lags behind that of pure Mamba (Gu & Dao, 2023). Based on Hyena, StripedHyena (Poli et al., 2023b) interleaves attention and SSM layers in a 7B parameter model. However, it lags behind the Attention-only Mistral-7B (Jiang et al., 2023). All of this renders Jamba the first production-grade Attention-SSM hybrid model.[1] Scaling the hybrid Jamba architecture required overcoming several obstacles, dicsussed in Section C.

Jamba also includes MoE layers (Shazeer et al., 2017; Fedus et al., 2022), which allow increasing the model capacity (total number of available parameters) without increasing compute requirements (number of active parameters). MoE is a flexible approach that enables training extremely large models with strong performance (Jiang et al., 2024). In Jamba, MoE is applied to some of the MLP layers (similar to Anthony et al. (2024), but they applied MoE to all MLPs). The more MoE layers, and the more experts in each MoE layer, the larger the total number of parameters. In contrast, the more experts we use at each forward pass, the more active parameters and the higher the compute.

To support cost-effective inference with large Jamba models, we introduce ExpertsInt8, a novel quantization technique that allows fitting Jamba-1.5-Large on a single machine with 8 80GB GPUs when processing texts with 256K-token contexts without loss of quality (Section 4).

We evaluated the Jamba models on a wide range of benchmarks and found they performs comparably to state-of-the-art open-weight models of a similar or greater number of parameterts, while offering much better throughput. Notably, our models support a context length of 256K tokens – the longest supported context length for production-grade publicly available models. Jamba-1.5 models shine at long-context evaluations, making them the only models with an effective length of 256K on the RULER benchmark (Hsieh et al., 2024), while offering 10x reduction in KV cache memory as well as superior throughput and latency.

We make the Jamba models publicly available under the Jamba Open Model License to support further study, experimentation, and optimization of this novel architecture by the community:
**Jamba-1.5-Mini:** https://huggingface.co/ai21labs/AI21-Jamba-1.5-Mini
**Jamba-1.5-Large:** https://huggingface.co/ai21labs/AI21-Jamba-1.5-Large

## 2 ARCHITECTURE

Jamba is a hybrid decoder architecture that mixes Transformer layers (Vaswani et al., 2017) with Mamba layers (Gu & Dao, 2023), a recent state-space model (SSM) (Gu et al., 2021b;a), in addition to a mixture-of-experts (MoE) module (Shazeer et al., 2017; Fedus et al., 2022). We call the combination of these three elements a Jamba block. See Figure 1 for an illustration.

---

[1]A couple of other attempts to scale hybrid models were announced after an earlier release of Jamba (Lieber et al., 2024), confirming our design choices (Dao & Gu, 2024; Waleffe et al., 2024); we discuss this more below. Other recent attempts include Samba (Ren et al., 2024), which combines Mamba with sliding-window attention, and YOCO (Sun et al., 2024), which introduces a decoder-decoder architecture.

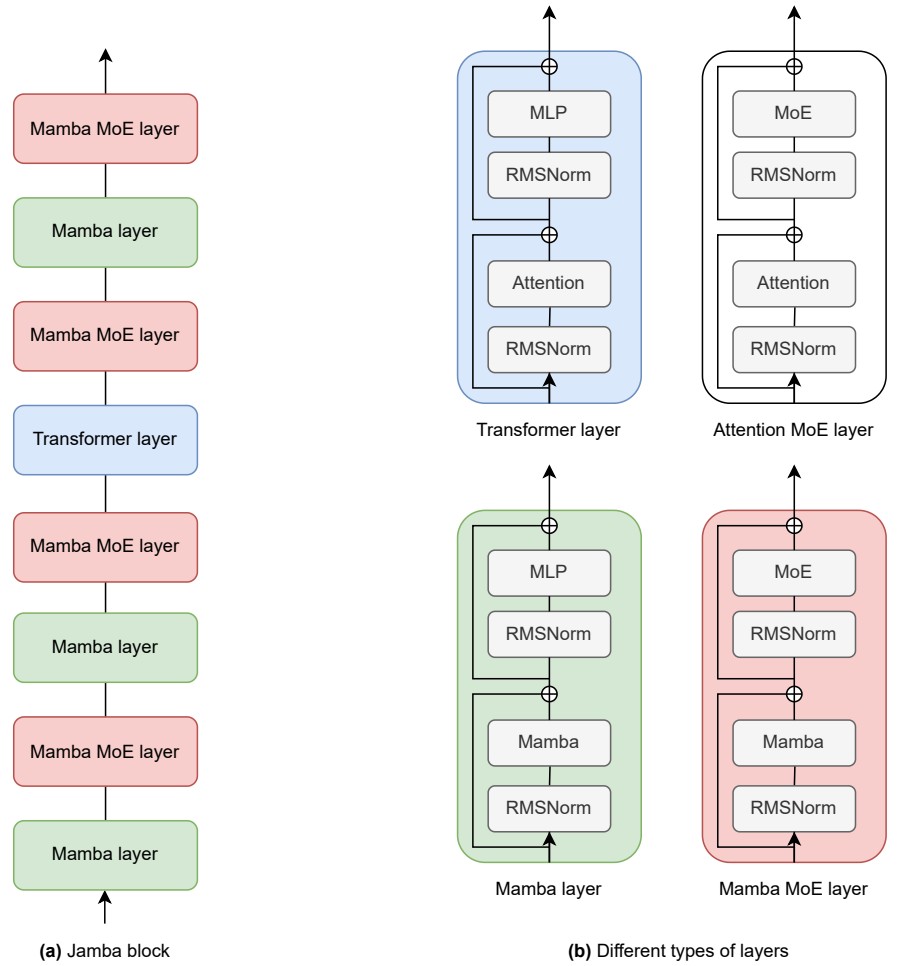

**(a)** Jamba block            **(b)** Different types of layers

Figure 1: **(a)** A single Jamba block. **(b)** Different types of layers. The implementation shown here is with $l = 8$, $a : m = 1 : 7$ ratio of attention-to-Mamba layers, and MoE applied every $e = 2$ layers.

Combining Transformer, Mamba, and MoE elements allows flexibility in balancing among the sometimes conflicting objectives of low memory usage, high throughput, and high quality. In terms of memory usage, comparing the total size of the model parameters can be misleading. In an MoE model, the number of active parameters that participate in any given forward step may be much smaller than the total number of parameters. Another important consideration is the KV cache—the memory required to store the attention keys and values in the context—which becomes a limiting factor when scaling Transformer models to long contexts. Trading off attention layers for Mamba layers reduces the total size of the KV cache. Our architecture provides not only a small number of active parameters but also an 8x smaller KV cache compared to a vanilla Transformer.

In terms of throughput, with short sequences, attention operations take up a small fraction of the inference and training FLOPS (Chowdhery et al., 2023). However, with long sequences, attention hogs most of the compute. In contrast, Mamba layers are more compute-efficient. Thus, increasing the ratio of Mamba layers improves throughput especially for long sequences.

Here is a description of the main configuration, which provides improved performance and efficiency. Section C contains results from ablation experiments supporting the design choices.

The basic component is a Jamba block, which may be repeated in sequence. Each block combines Mamba and Attention layers. Each such layer contains either an attention or a Mamba module, followed by an MLP. The different possible types of layers are shown in Figure 1b.[2] A Jamba block contains $l$ layers, mixed at a ratio of $a : m$, meaning $a$ attention layers for every $m$ Mamba layers.

---

[2] The figure shows a potential Attention MoE layer, which our Jamba does not use, but future variants could.

In Jamba, some of the MLPs may be replaced by MoE layers, which increase model capacity while keeping the active number of parameters, and thus the compute, small. The MoE may be applied to MLPs every $e$ layers. When using MoE, there are $n$ possible experts per layer; a router chooses the top-$K$ experts at each token. In summary, the different degrees of freedom in the Jamba architecture are:

- $l$: The number of layers.
- $a : m$: ratio of attention-to-Mamba layers.
- $e$: how often to use MoE instead of a single MLP.
- $n$: total number of experts per layer.
- $K$: number of top experts used at each token.

Given this design space, Jamba provides flexibility in preferring certain properties over others. For instance, increasing the ratio of Mamba layers ($m$) at the expense of attention layers ($a$), reduces the required memory for storing the KV cache. This reduces the overall memory footprint, which is especially important for processing long sequences. Increasing the ratio of Mamba layers also improves throughput, especially at long sequences. However, decreasing $a$ might harm model capabilities.

Additionally, balancing $n$, $K$, and $e$ affects the relationship between active parameters and total available parameters. A larger $n$ increases model capacity at the expense of memory footprint, while a larger $K$ increases the active parameter usage and the compute requirement. In contrast, a larger $e$ decreases the model capacity, while decreasing both compute (when $K > 1$) and memory requirements, and allowing for fewer communication dependencies (decreasing memory transfers as well as inter-GPU communication during expert-parallel training and inference).

Jamba's implementation of Mamba layers incorporate several normalizations that help stabilize training at large scales. In particular, we apply RMSNorm (Zhang & Sennrich, 2019) in the Mamba layers.

We experimented also with Mamba-2 (Dao & Gu, 2024), a faster and improved version of Mamba, which was reported to outperform Mamba and Transformers separately. However, as shown in Appendix C.2, we found that in a hybrid architecture, the Mamba-1-Attention combination works better than Mamba-2-Attention, so we use Mamba-1 in the released models.

We found that with the Mamba layer, positional embeddings or mechanisms like RoPE (Su et al., 2024) are not necessary (Appendix C), and so we do not use any explicit positional information. Other architecture details are standard, and are detailed in Appendix B.

## 3 MODEL CONFIGURATION

The specific configurations in our implementation were chosen to fit in a single 80GB GPU (for Jamba-1.5-Mini) or a single 8x80GB GPU machine (for Jamba-1.5-Large), while achieving best performance in the sense of quality and throughput. It is based on the following main insights, which stem from ablations experiments (Appendix C):

- Hybrid Attention-Mamba performs better than vanilla Attention or Mamba models.
- There is no substantial difference in performance between 1:3 and 1:7 ratios of Attention-to-Mamba layers, while 1:7 is more compute-efficient.
- Pure Mamba struggles with in-context learning (ICL) while the hybrid Attention-Mamba succeeds in ICL and acquires induction heads (Olsson et al., 2022).
- MoE improves on hybrid Attention-Mamba at large scale.

Concretely, our implementation of Jamba uses a sequence of Jamba blocks (4 blocks in Jamba-1.5-Mini, 9 in Jamba-1.5-Large). Each block has the following configuration:

| | | | | |
|---|---|---|---|---|
| Number of layers ($l$) | 8 | Total number of experts ($n$) | 16 |
| Ratio of attention-to-Mamba layers ($a : m$) | $1 : 7$ | # top experts used at each token ($K$) | 2 |
| Use MoE instead of MLP every $e$ layers | 2 | | |

The MoE configuration was chosen to enable the model to fit in a single 80GB GPU (or an 8-GPU machine for Jamba-1.5-Large, with appropriate quantization; Section 4.1), while including enough memory for the inputs. Thus, $n$ and $e$ were balanced to have an average of ∼8 experts per layer. We also balanced $n$, $K$, and $e$ to allow for high quality, while reducing both compute requirements and communication dependencies (memory transfers). Thus, we replace the MLP module with MoE on every other layer, and we have a total of 16 experts, with two used at each token. These choices were inspired by prior work on MoE (Zoph et al., 2022; Clark et al., 2022) and verified in preliminary experiments.

Table 1: Comparison of Jamba-1.5-Mini, Jamba-1.5-Large. and recent open models in terms of total available parameters, active parameters, and KV cache memory on long contexts. Jamba-1.5-Mini and Jamba-1.5-Large provide substantial reductions in the KV cache memory requirements.

|  | Available params | Active params | KV cache (256K context, 16bit) |
|---|---|---|---|
| Mistral | 7.2B | 7.2B | 32GB |
| Mixtral 8x7B | 46.7B | 12.9B | 32GB |
| Llama-3.1 8B | 8B | 8B | 32GB |
| Mixtral 8x22B | 141B | 39B | 56GB |
| Mistral-Large-2 | 123B | 123B | 88GB |
| Llama-3.1 70B | 70B | 70B | 80GB |
| Llama-3.1 405B | 405B | 405B | 252GB |
| **Jamba-1.5-Mini** | 52B | 12B | 4GB |
| **Jamba-1.5-Large** | 398B | 94B | 9GB |

Table 1 compares the Jamba-1.5 models to publicly available models of similar sizes. Jamba-1.5-Mini has a similar number of active parameters as Mixtral 8x7B, while Jamba-1.5-Large's active parameter count is between Llama-3.1-70B and Mistral-Large-2. At the same time, both our Jamba models have a much smaller KV cache memory usage (at 256K tokens) compared to all other models, with roughly an order of magnitude reduction compared to their respective counterparts.

With these settings, and our specialized quantization (Section 4.1), Jamba-1.5-Large can be served on a single machine with 8 80GB GPUs with context lengths up to 256K tokens.

## 4 SERVING CONSIDERATIONS AND IMPROVEMENTS

We share a few insights and improvements that allow for efficient serving of large Jamba models.

### 4.1 EXPERTSINT8 QUANTIZATION

To support efficient serving of Jamba-1.5-Large, we developed a new quantization technique, which we dub ExpertsInt8. We observe that over 85% of the model weights are in the MoE layers, and over 90% are in MoE or MLP layers. We wish to quantize these weights while still enjoying the benefits of fast BF16 kernels. To do so, we quantize the MoE and MLP weights to INT8, save them in INT8, and dequantize them back to BF16 before the actual computation. Importantly, the dequantization step happens directly inside the `fused_moe` kernel in vLLM (Kwon et al., 2023). In this way, the dequantization process adds negligible overhead, and even leads to improved latency over BF16.[3] We have contributed our modified `fused_moe` kernel to vLLM.[4]

Our ExpertsInt8 method has several advantages. First, it is fast; quantization only takes a few seconds at model loading. Second, unlike most other techniques in vLLM, it does not rely on calibration, which can take hours or days and be unstable. Third, we can still use BF16 to hold large activations. Fourth, it is available to use on A100 GPUs, unlike FP8, which is only available on H100. Finally, our quantization matches FP8 in latency, while surpassing other quantization techniques, without a loss in quality (Appendix C.7).

Figure 2 compares the latency with different quantization techniques using Jamba-1.5-Mini, Jamba-1.5-Large, and two Mixtral models (8x78B and 8x22B). On H100 GPUs, ExpertsInt8 matches the latency of FP8. On A100, where FP8 is unavailable, ExpertsInt8 shines, outperforming GPTQ (Frantar et al., 2023) by a large margin. Finally, ExpertsInt8 comes with no noticeable degradation in quality (Appendix C.7). Together with the advantages of ExpertsInt8 explained above, this makes it an attractive quantization technique for serving large MoE models.[5]

---

[3]We attribute this to the kernel operating on relatively small blocks of weights and activations, which it moves from GPU HBM to SRAM prior to performing the computations. In our implementation, the weights move from HBM to SRAM when they are in int8, so it takes less time as their memory footprint is cut by half.

[4]https://github.com/vllm-project/vllm/pull/7415

[5]Other MoE quantization attempts are Kim et al. (2023) and the recent work of Li et al. (2024a), both focused on very low-bit quantization, which downgrades performance.

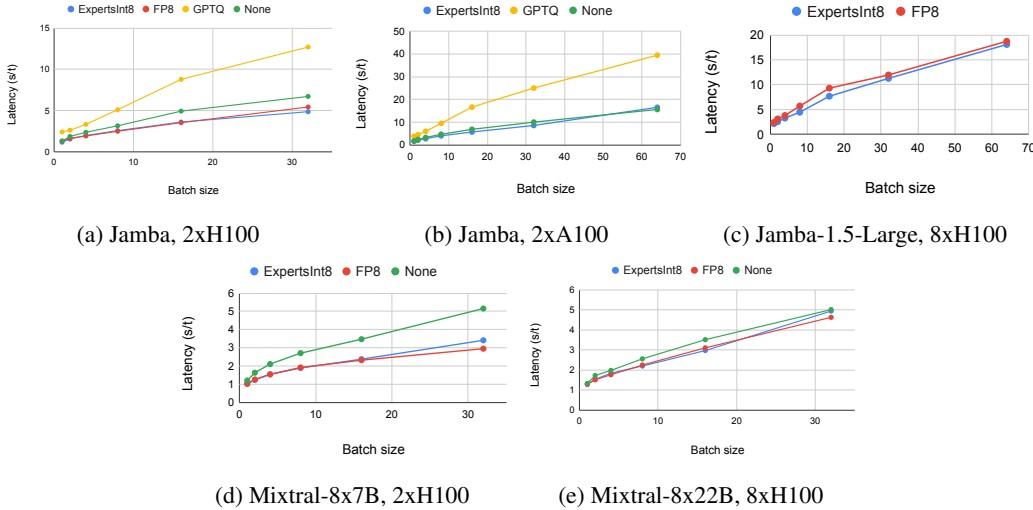

Figure 2: Comparison of different quantization techniques. End-to-end latency with 1024-token context and 128-token decoding. ExpertsInt8 performs similar to FP8, while being fast and simple to apply, and allowing BF16 activations, as well as applicable to A100 GPUs, where FP8 is unavailable.

## 4.2 ACTIVATION LOSS

During pre-training, we found that certain activations, namely outputs of specific experts and the last Mamba layers, were gradually increasing in magnitude for certain input tokens, reaching values as high as $4 \times 10^6$. Although this did not hurt the pre-training itself, which was done in BF16 precision, the activations' magnitude could cause numerical issues during inference as some quantization libraries support only FP16 precision for activations, which has a maximum range of 64K. To alleviate these concerns, we added an "Activation Loss" term, proportional to the mean-square of activations, with a configurable $\alpha$ factor, which penalizes larger activation values. We found via experimentation that this auxilary loss has no affect on training even with $\alpha$ values up to at least $10^{-3}$. For Jamba-1.5-Large, we used $\alpha = 10^{-5}$, which reduced the activations to an acceptable range (2K-3K max). Moreover, adding this auxilary loss reduced the activations almost instantly, allowing it to be added only towards the end of training without any affect on training speed and quality.

To validate this approach, we ran our full evaluation suite on the model using FP16 activations and obtained the same results as the BF16 evaluations without any nans/overflows.

## 5 THROUGHPUT AND LATENCY ANALYSIS

Thanks to the hybrid architecture, our Jamba-1.5 models provide excellent throughput and latency. As Figures 3 and 4 show, our models obtain much better latency and throughput than similarly-sized models. Their advantage shines at long contexts, with substantial gaps. Importantly, Jamba-1.5-Large runs efficiently even at long contexts, where the large LLaMA3-405B cannot run on the same hardware.

## 6 TRAINING

### 6.1 TRAINING INFRASTRUCTURE AND DATA

Jamba-1.5-Large was trained on NVIDIA H100 GPUs using our in-house proprietary framework, which includes FSDP, tensor parallelism, sequence parallelism, and expert parallelism. For the latter we have adapted MegaBlocks (Gale et al., 2023).

### 6.2 TRAINING STAGES

The model was trained in three stages. During pre-training, it was first trained on an in-house dataset last updated in March 2024. Our pre-training dataset is a mixture of publicly available web documents, code, books, scientific articles, and Wikipedia. Our pre-processing pipeline includes parsing, quality filters, and deduplication. To make the best use of publicly available data, we developed our own in-house parser, and used it to extract text and formatting. The exact data mixture was determined through various ablations. This stage included multilingual data with emphasis on the following

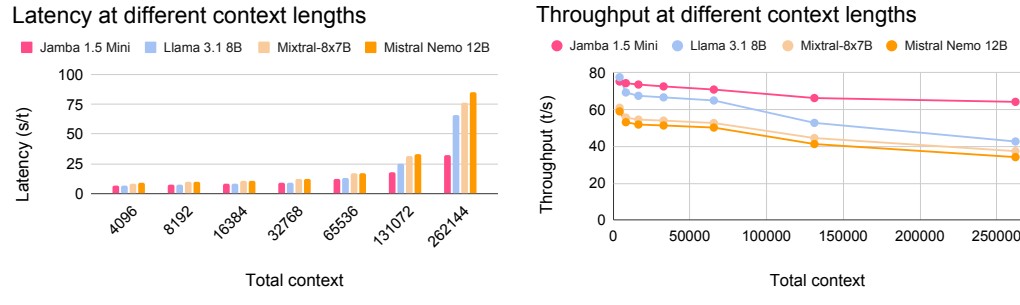

(a) Jamba-1.5-Mini, end-to-end latency.     (b) Jamba-1.5-Mini, output tokens throughput.

Figure 3: Comparison of Jamba-1.5-Mini to other models in terms of latency and throughout. All measurements are on 2xA100 80GB GPUs, with batch size 1 and output length 512 tokens. Jamba-1.5-Mini exhibits better latency, especially at large contexts, with only a slight reduction in throughput.

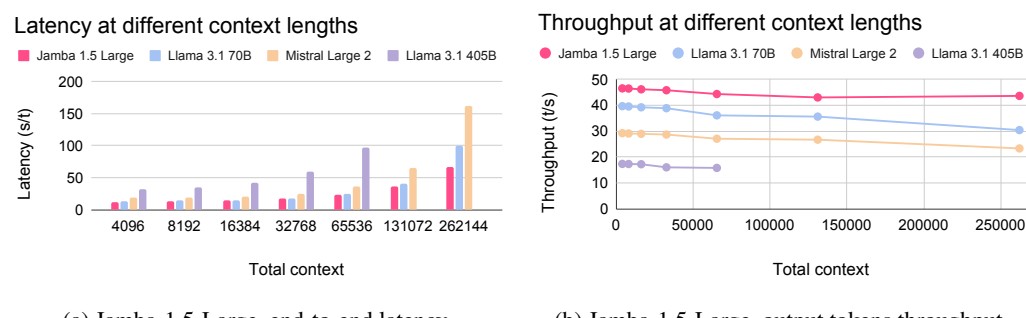

(a) Jamba-1.5-Large, end-to-end latency.     (b) Jamba-1.5-Large, output tokens throughput.

Figure 4: Comparing Jamba-1.5-Large to other models in terms of latency and throughout. All measurements are on 8xA100 80GB GPUs, with batch size 1 and output length 512 tokens. Jamba-1.5-Large exhibits better latency, especially at long contexts, with only a slight reduction in throughput. Llama-3.1-405B results stop at 64K as it cannot fit contexts over $\sim 100K$ tokens on 8 80GB GPUs.

languages: English, Spanish, French, Portueguse, Italian, Dutch, German, Arabic, and Hebrew. The pre-training context length was 4K for Jamba-1.5-Mini and 8K for Jamba-1.5-Large. It was then trained for a short phase of mid-training with a high proportion of long documents to emphasize its long-range capabilities. Finally, the model went through post-training, described in the next section.

## 6.3 POST-TRAINING

Our approach to post-training aims to achieve two objectives simultaneously: (*i*) provide the model with various skills and conversational capabilities; (*ii*) retain capabilities from pre-training and especially the long-context capabilities from mid-training. These two objectives are partly conflicting, since most of the available post-training datasets consist of relatively short examples.

Given these considerations, our post-training process involves supervised fine-tuning (Wei et al., 2022; Sanh et al., 2022) on high-quality conversational data, skill-specific data, and long-context data. Mixing these different types of data aims to retain long-context capabilities and acquire desired skills. As shown in the evaluations below, we find that our models perform very well in long-context evaluations.

In supervised fine-tuning, we use three kinds of datasets: general instruction-following, code and math instructions, and specific skills. For the latter, we make heavy use of synthetic data, as common in recent foundation models (Dubey et al., 2024) and reflecting our approach for constructing structured data for building compound AI systems (Lenz et al., 2024). We develop multiple data synthesis pipelines, targeting different capabilities, which apply the following pattern: (i) Sample or generate prompts in a target distribution; (ii) Generate responses from language models; (iii) Filter or rank responses by quality using automatic validation and scoring; and (iv) Post-edit to remove artifacts and fit desired formatting. We use different models, prompting, sampling, filtering, and editing for different pipelines that compose the final data mixes. Appendix D provides more details and examples for synthetic data generation processes, as well as several observations from our post-training work.

## 7  EVALUATION

This section performs a comprehensive evaluation of Jamba-1.5 models. First, we report results on long-context settings, where the advantages of the Jamba arhcitecture are most pronounced. Then, we show that our models are competitive in on standard academic benchmarks, as well as chatbot benchmarks. We conclude with a small multilingual evaluation.

We mainly compare with recent open-weight models of the same size range: Llama-3.1 70B and Mistral-Large-2-123B when comparing with Jamba-1.5-Large; Llama-3.1-8B and Gemma-2-9B when comparing with Jamba-1.5-Mini.

### 7.1  LONG-CONTEXT EVALUATIONS

The benefits of Jamba models shine in long contexts, where they offer superior throughput (Section 5). This section evaluates them on synthetic and naturalistic long-context benchmarks.

#### 7.1.1  RULER

We evaluate on the RULER benchmark (Hsieh et al., 2024), a set of 13 synthetic tasks aimed to assess long-context capabilities of language models. RULER includes 8 variants of needle-in-a-haystack retrieval tasks (Kamradt, 2023; Mohtashami & Jaggi, 2023; Li et al., 2023; Liu et al., 2023), including multiple 'needles' (Arora et al.). It also has one variable tracking task, two aggregation tasks where one should return the most common words, and two question-answering tasks, where paragraphs cotraining answers from naturalistic datasets (Rajpurkar et al., 2018; Yang et al., 2018) are inserted into random paragraphs to simulate long contexts.

The results for key proprietary and open-weight models are shown in Table 2; full results are given in Appendix E.1. Among all publicly available and proprietary models, Jamba-1.5-Mini and Jamba-1.5-Large are the only ones with a confirmed effective length of 256K tokens.[6]

Table 2: Comparison of Jamba-1.5 models with other publicly available and proprietary models on the RULER benchmark. Results for other models are from the RULER Github. Jamba-1.5 models are the only ones with a confirmed effective length of 256K tokens.

|  | Claimed Length | Effective Length | 4k | 8k | 16k | 32k | 64k | 128k | 256k | Avg. |
|---|---|---|---|---|---|---|---|---|---|---|
| **Jamba-1.5-Large** | 256K | **256K** | 96.7 | 96.6 | 96.4 | 96.0 | 95.4 | 95.1 | 93.9 | 95.7 |
| **Jamba-1.5-Mini** | 256K | **256K** | 95.7 | 95.2 | 94.7 | 93.8 | 92.7 | 89.8 | 86.1 | 92.6 |
| Gemini-1.5-pro | 1M | >128K | 96.7 | 95.8 | 96 | 95.9 | 95.9 | 94.4 | - | 91.4 |
| GPT-4-1106-preview | 128K | 64K | 96.6 | 96.3 | 95.2 | 93.2 | 87 | 81.2 | - | 91.6 |
| Llama 3.1 70B | 128K | 64K | 96.5 | 95.8 | 95.4 | 94.8 | 88.4 | 66.6 | - | 89.6 |
| Llama 3.1 8B | 128K | 32K | 95.5 | 93.8 | 91.6 | 87.4 | 84.7 | 77 | - | 88.3 |
| Mistral Large 2 | 128K | 32K | 96.2 | 96.1 | 95.1 | 93 | 78.8 | 23.7 | - | 80.5 |

#### 7.1.2  ∞BENCH

Next we evaluate on ∞BENCH (Zhang et al., 2024), which has an average length of 100K tokens. We focus on two English tasks on understanding long novels: question answering (EN.QA) and multiple-choice question answering (EN.MC). As Table 3 shows, Jamba-1.5 models perform very well in this case, outperforming similarly sized Llama-3.1 and Mistral-Large-2 models. (We do not report results with Gemma-2 9B due to its short context window of 8K.)

Table 3: Jamba-1.5 models outperform similarly sized Llama-3 and Mistral-Large-2 models in long-context evaluations. † evaluation run by us.

| Benchmark | **Jamba-1.5 Mini** | Llama-3.1 8B | Gemma-2 9B | **Jamba-1.5 Large** | Llama-3.1 70B | Mistral-L-2 123B |
|---|---|---|---|---|---|---|
| EN.MC | 76.9 | 65.1 | - | 80.4 | 78.2 | 36.9† |
| EN.QA | 40.6 | 27.1 | - | 34.9 | 36.7 | 15.4† |

[6]In the official results, Gemini-pro has effective length >128K but RULER did not confirm longer contexts.

## 7.2 ACADEMIC BENCHMARKS

While not our main focus, we report results on a range of standard academic benchmarks: MMLU (Hendrycks et al., 2020), MMLU-Pro (Wang et al., 2024), GPQA (Rein et al., 2023), ARC-Challenge (Clark et al., 2018), BBH (Suzgun et al., 2023), and HumanEval (Chen et al., 2021). We also evaluate on the IFEval instruction following dataset (Zhou et al., 2023) and the BFCL v1 function calling dataset (Yan et al., 2024). Finally, we report safety evaluations on RealToxicity (Gehman et al., 2020) and TruthfulQA (Lin et al., 2022).

Table 4 compares Jamba-1.5-Large to several publicly available models at similar sizes. All results are either taken from official sources or evaluated by us, as indicated in the table.[7] We observe that the Jamba-1.5 models perform similarly to recent state-of-the-art publicly available models on standard academic benchmarks, including knowledge, reasoning, instruction following and function calling capabilities. We also observe similar safety metrics as those reported in the literature.

Importantly, the Jamba-1.5 models achieve these results while providing much better throughput and latency, as discussed above.

Table 4: Jamba-1.5 models obtain similar performance to similarly sized models while enjoying a better throughput and latency. $^\dagger$ evaluation run by us. $^\diamond$ reported in the HuggingFace OpenLLM leaderboard. $^\ddagger$ Lacking function calling capabilities. $^\star$ Strict/flexible evaluation.

| Benchmark | Metric | Jamba-1.5 Mini | Llama-3.1 8B | Gemma-2 9B | Jamba-1.5 Large | Llama-3.1 70B | Mistral-L-2 123B |
|---|---|---|---|---|---|---|---|
| MMLU | 5-shot | 69.7 | 69.4 | 71.3 | 80.0 | 83.6 | 82.5$^\dagger$ |
| MMLU Pro | 5-shot | 39.8 | 38.0$^\diamond$ | 39.0$^\diamond$ | 48.3 | 53.0$^\diamond$ | 54.2$^\dagger$ |
| GPQA | 0-shot | 32.3 | 27.0$^\diamond$ | 36.0$^\diamond$ | 36.9 | 36.0$^\diamond$ | 40.7$^\dagger$ |
| ARC-C | 0-shot | 85.7 | 83.4 | 68.4 | 93.0 | 94.8 | 65.0$^\dagger$ |
| BBH | 3-shot | 53.4 | 51.0$^\diamond$ | 60.0$^\diamond$ | 65.5 | 69 | 70.8$^\dagger$ |
| HumanEval | pass@1 | 62.8 | 72.6 | 40.2 | 71.3 | 80.5 | 92 |
| GSM8K | 5-shot | 75.8 | 75.2/83.7$^\star$ | 68.6 | 87.0 | 71.5/94.2$^\star$ | 91.0$^\dagger$ |
| IFEval | 0-shot | 75.8 | 80.4 | 74.3 | 81.5 | 87.5 | 87.8$^\dagger$ |
| BFCL | 0-shot | 80.7 | 76.1 | -$^\ddagger$ | 85.5 | 84.8 | 85.1$^\dagger$ |
| RealToxicity | avg tox | 8.1 | - | 8.2 | 6.7 | - | - |
| TruthfulQA | 0-shot | 54.1 | 51.5$^\dagger$ | 50.2 | 58.3 | 60.7$^\dagger$ | 50.4$^\dagger$ |

## 7.3 CHATBOT EVALUATIONS

In this section we evaluate the Jamba-1.5 models on two chatbot scenarios: Arena-Hard (Li et al., 2024b), a set of 500 challenging user queries that uses GPT4-Turbo as a judge, and WildBench (Lin et al., 2024), which uses GPT4-Turbo as a judge with a length bias mitigation. We then report results from live interactions with users on the Chatbot Arena. As Table 5 shows, Jamba-1.5 models obtain excellent results in these evaluations, with Jamba-1.5-Large surpassing Llama-3.1 70B, but somewhat trailing behind Mistral-Large-2 123B, which has about 30% more active parameters.

Table 5: Comparison of Jamba-1.5 models to similarly sized models on chatbot benchmarks. Jamba-1.5 models obtain similar performance with better throughput and latency. $^\dagger$evaluation run by us.

| Benchmark | Jamba-1.5 Mini | Llama-3.1 8B | Gemma-2 9B | Jamba-1.5 Large | Llama-3.1 70B | Mistral-L-2 123B |
|---|---|---|---|---|---|---|
| Arena-Hard | 46.1 | 21.3 | 43.2$^\dagger$ | 65.4 | 55.7 | 70.4 |
| Wild-Bench | 42.4 | 33.6$^\dagger$ | 42.7 | 48.5 | 49.8$^\dagger$ | 56.3$^\dagger$ |

---

[7]In two cases we failed to obtain good results: Mistral-Large-2 fails to obtain good scores on ARC-C despite multiple attempts. Llama-3.1 models perform poorly on GSM8K with the standard strict evaluation mode, so we also report for them a flexible evaluation, which allows higher results.

Figure 5 provides ELO scores from the Chatbot Arena (as of September 10, 2024), where humans make live pairwise comparisons of models. Jamba models perform comparably to state-of-the-art open-weight models of similar size, while enjoying the efficiency benefits of the Jamba architecture.

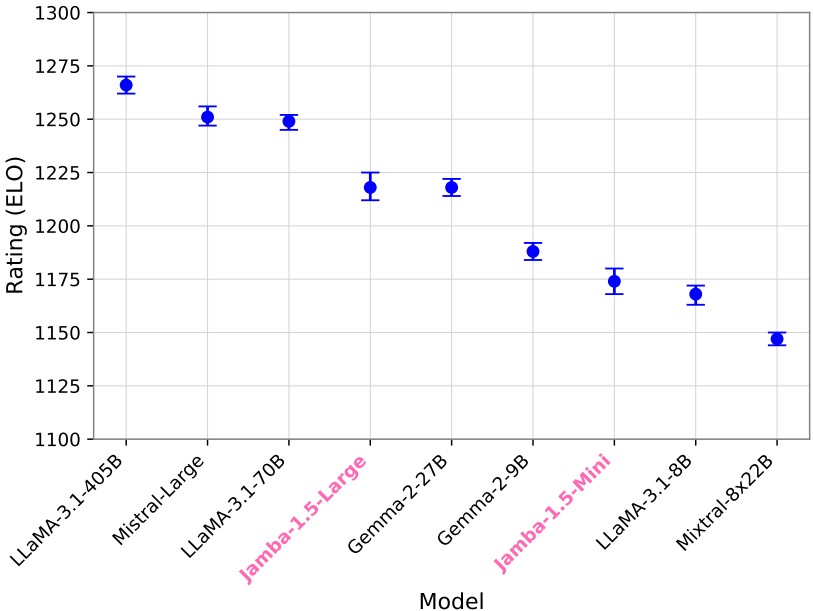

Figure 5: Chatbot Arena ELO scores of Jamba with open-weight models of a similar size. Jamba models obtain good performance while enjoying a better throughput and latency.

## 7.4 MULTILINGUAL CAPABILITIES

We perform a basic evaluation of Jamba-1.5 abilities in non-English languages on the multilingual MMLU dataset (Lai et al., 2023), available via the LM Evaluation Harness (Gao et al., 2024). As Table 6 shows, Jamba-1.5-Mini performs similarly or better than its points of comparison. Jamba-1.5-Large is slightly behind its comparable models, but still exhibits good multilingual capabilities.

Table 6: Comparison of Jamba-1.5 with other models on the multilingual MMLU dataset.

|  | Spanish | Portuguese | French | German | Arabic | Italian | Dutch | Avg |
|---|---|---|---|---|---|---|---|---|
| **Jamba-1.5-Mini** | 66.3 | 66.7 | 65.9 | 63.8 | 57.3 | 65.1 | 65.0 | 64.30 |
| Llama-3.1-8B | 59.5 | 59.1 | 59.5 | 57.2 | 46.9 | 58.4 | 57.2 | 56.83 |
| Gemma-9B | 66.0 | 59.9 | 66.7 | 64.3 | 55.9 | 65.8 | 64.8 | 63.34 |
| **Jamba-1.5-Large** | 75.5 | 75.5 | 75.8 | 73.9 | 67.1 | 75.2 | 74.6 | 73.94 |
| Llama-3.1-70B | 79.5 | 79.4 | 79.1 | 78.4 | 70.4 | 79.1 | 78.4 | 77.76 |
| Mistral-Large-2 | 78.7 | 78.4 | 78.4 | 77.4 | 65.9 | 78.3 | 76.2 | 76.19 |

## 8 CONCLUSION

We presented Jamba, a novel architecture that combines Attention and Mamba layers, with MoE modules, and an open implementation of it in two sizes: Jamba-1.5-Mini with 12B active and 52B total parameters and Jamba-1.5-Large with 94B active and 398B total parameters. We showed how Jamba offers flexibility for balancing performance and memory needs, while maintaining a high throughput. We experimented with several design choices such as the ratio of Attention-to-Mamba layers and discussed some discoveries made during the development process, which will inform future work on hybrid models. We also developed ExpertsInt8, a novel quantization to support cost-effective serving of such models at a large scale, even on a single machine with long contexts. Both models achieve excellent performance in long-context evaluations, academic benchmarks, and chatbot evaluations, while offering improved latency and throughput, especially for long contexts. We release the model weights for use by the community in hopes that others build on this technology.

CONTRIBUTIONS

**Pre- and Post-Training**
Alan Arazi
Barak Lenz[*]
Chen Almagor
Dan Padnos[*]
Daniel Gissin[*]
Daniel Jannai
Dor Muhlgay
Edden M Gerber
Erez Safahi
Gal Cohen
Gal Shachaf
Hofit Bata
Inbal Magar
Itay Dalmedigos
Jhonathan Osin[*]
Matan Danos
Michael Gokhman
Nir Ratner
Noam Gat
Noam Rozen
Omer Antverg
Omri Abend
Opher Lieber[*]
Orit Cohavi
Raz Alon
Shai Shalev-Shwartz
Shaked Meirom
Tom Braude
Uriya Pumerantz
Yonatan Belinkov
Yuval Globerson
Yuval Peleg Levy

**Serving & Infrastructure**
Amir Bergman
Avshalom Manevich
Barak Peleg
Elad Dolev
Eran Krakovsky
Erez Schwartz
Haim Rozenblum
Mor Zusman
Oded Fried
Roman Glozman
Shahar Lev
Tomer Asida
Yehoshua Cohen

**Data**
Ben Aviram
Dor Zimberg
Ido Blass
Ohad Leshno
Rom Gilad
Tom Ben Gal

**Evaluation**
Clara Fridman
Julie Fadlon
Maria Rozman
Naama Gidron
Ro'i Belson
Tal Ness

**Project & Product Management**
Or Dagan[*]
Roi Cohen
Shaked Meirom[*]
Tal Delbari
Yoav Shoham

---

[*]Project leads

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

## A    LIMITATIONS

One limitation of this work is that some details about the training process are not disclosed, such as the total training budget in terms of tokens and compute, and the data sources for pre-training and post-training. However, we provide ablations of the Jamba architecture with 1.3B/7B parameter models trained for 250B/50B tokens with the exact same data and compute budget, which reveal insights about the novel aspects of the Jamba architecture at a fairly large scale.

## B    ARCHITECTURE AND IMPLEMENTATION DETAILS

The main features of the Jamba architecture are explained in Section 2, with ablations supporting our design choices in Appendix C.

Other details are standard in recent language models. Concretely, we use grouped-query attention (GQA), SwiGLU activation function (Shazeer, 2020; Chowdhery et al., 2023; Touvron et al., 2023), load-balancing loss to balance the routing decisions in the MoE and a router z-loss to avoid large router logits (Fedus et al., 2022; Zoph et al., 2022). The vocabulary size is 64K. The tokenizer is trained with SentencePiece BPE (Gage, 1994; Sennrich et al., 2016; Mielke et al., 2021) and each digit is a separate token (Chowdhery et al., 2023). We also remove the dummy space used in Llama and Mistral tokenizers for more consistent and reversible tokenization.

We have made sure to use the same evaluation setup in all models compared in this work, including prompts, official splits, and number of shots. When evaluating Jamba and self-reporting results of other models, we always used the official repositories. We mainly used the LM Evaluation Harness (Biderman et al., 2024) whenever possible. Regarding evaluations not in LM Harness: For HumanEval, BFCL, and IFEval, we used the official dataset and metrics, as mentioned in their official repos. For RealToxicity, we used the official dataset and used their API for judgment.

To evaluate the efficiency of Mamba and Attention layers during pre-training, we measured the models FLOPs utilization metric (MFU), proposed by Chowdhery et al. (2023). In our measurements, the MFU with vanilla attention drops from 60% to 20% at 32K token length, while the Mamba MFU remains close to 60%. In longer contexts, the difference would be even larger.

## C    ABLATIONS AND INSIGHTS

This section discusses ablation experiments we ran for different design choices in our implementation of the Jamba architecture. First we show the benefit of combining attention and Mamba layers, at which ratio they should be combined, and how to interleave them.[*] We investigate cases where pure Mamba fails, suggesting that it struggles to develop in-context learning capabilities, while the Attention-Mamba hybrid exhibits in-context learning similar to vanilla Transformers. Then we show the benefit of adding MoE on top of a hybrid Attention-Mamba model. Finally, we share two additional learnings that we found useful: explicit positional information is not needed in Jamba, and Mamba layers necessitate special normalization to stabilize training at large scale.[*]

For these ablations, we report the following measures, which exhibit informative performance even at small data or model scale. (In Appendix C.8 we report ablations of the architecture in terms of latency.)

- Academic benchmarks: HellaSwag (10-shot) (Zellers et al., 2019), WinoGrande (5-shot) (Sakaguchi et al., 2020), Natural Questions closed-book (NQ; 5-shot) (Kwiatkowski et al., 2019), ARC-Challenge (25-shot) (), BoolQ (10-shots) (Clark et al., 2019), IMDB (Maas et al., 2011), and NarrativeQA (NQA; Kocisky et al., 2018).
- HuggingFace OpenLLM leaderboard (OLLM) (Face, 2024): a summary statistic of several datasets. We report results with our reproduction.
- Perplexity evaluations: we report log-prob (per byte) on texts from three domains: C4, Books, and code.

---

[*]Our design choices were later confirmed by independent experiments of others (Dao & Gu, 2024; Waleffe et al., 2024), which followed an earlier Jamba release.

[*]In all the ablations, "pure Mamba" refers to models with Mamba layers interleaved with MLP layers.

## C.1 BENEFITS OF COMBINING ATTENTION AND MAMBA

We first investigate the ratio of Attention-to-Mamba layers ($a : m$), with 1.3B parameters models trained for 250B tokens. As Table 7 shows, the hybrid Jamba model outperforms the pure attention or Mamba models. The ratio of attention-to-Mamba layers may be 1:3 or 1:7 with virtually no performance difference. Figure 6 shows the training loss of these models, where Jamba exhibits improved loss during training. Given that a 1:7 ratio is more compute-efficient and shows similar performance, we opt for it in our larger-scale experiments.

Table 7: Results on academic benchmarks and log-probability evaluations showing an improved performance of hybrid Attention-Mamba (no MoE) compared to vanilla Attention and Mamba models. There is no substantial difference between 1:3 and 1:7 ratios of Attention-to-Mamba layers. Models are 1.3B parameters, trained for 250B tokens.

| | OLLM | Hella Swag | Wino Grande | NQ | BoolQ | NQA | ARC-C | IMDB | log-prob | | |
| --- | --- | --- | --- | --- | --- | --- | --- | --- | --- | --- | --- |
| | | | | | | | | | C4 | Books | Code |
| Attention | 36.4 | 62.4 | 59.6 | 14.5 | 60.9 | 45.8 | 34.6 | 84.1 | -0.543 | -0.659 | -0.331 |
| Mamba | 36.1 | 62.6 | 59.4 | 14.5 | 61.1 | 27.7 | 34.1 | 48.8 | -0.543 | -0.661 | -0.334 |
| Hybrid, $1 : 3$ | 37.2 | 65.1 | 61.7 | 16.5 | 60.6 | 44.3 | 36.8 | 90.1 | -0.533 | -0.649 | -0.321 |
| Hybrid, $1 : 7$ | 37.2 | 65.1 | 61.7 | 16.0 | 64.4 | 41.9 | 34.8 | 90.5 | -0.533 | -0.650 | -0.321 |

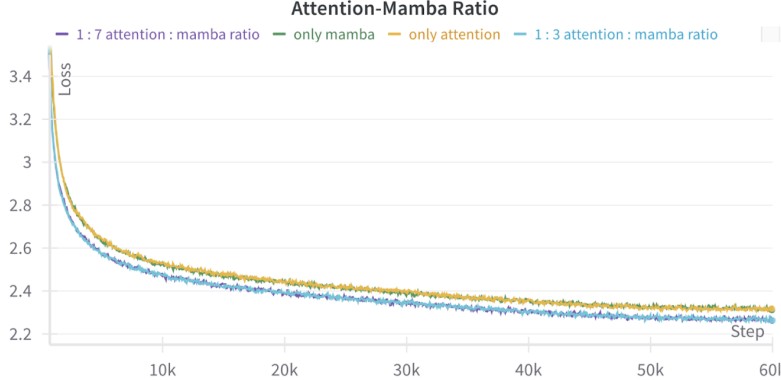

Figure 6: Training loss curves for pure Attention, pure Mamba, and Attention-Mamba hybrids (no MoE), with ratios $a : m$ of 1:3 and 1:7. All models are 1.3B parameters. The two hybrids achieve better loss throughout this training run, without any noticeable difference between the different Attention/Mamba ratios.

Next, we compare performance of vanilla Transformer, vanilla Mamba, and Attention-Mamba hybrid models, at 7B model size, after training on 50B tokens. As Table 8 shows, the pure Mamba model is quite competitive, but lags slightly behind pure Attention. The hybrid Attention-Mamba (without MoE) outperforms the pure models while obtaining better throughput than vanilla Transformers (Section 5).

Table 8: Results on academic benchmarks and log-prob evaluations, comparing pure Attention, pure Mamba, and Attention-Mamba hybrid ($a : m = 1 : 7$, no MoE). Models are 7B parameters, trained for 50B tokens.

| | OLLM | Hella Swag | Wino Grande | NQ | BoolQ | NQA | ARC-C | IMDB | log-prob | | |
| --- | --- | --- | --- | --- | --- | --- | --- | --- | --- | --- | --- |
| | | | | | | | | | C4 | Books | Code |
| Attention | 36.1 | 60.4 | 59.7 | 13.7 | 60.6 | 40.0 | 32.4 | 87.6 | -0.555 | -0.666 | -0.347 |
| Mamba | 35.3 | 60.2 | 55.8 | 14.0 | 52.0 | 19.4 | 33.8 | 64.0 | -0.554 | -0.667 | -0.355 |
| Hybrid | 36.6 | 62.5 | 58.8 | 15.4 | 59.4 | 38.3 | 35.0 | 87.1 | -0.547 | -0.658 | -0.340 |

Figure 7 shows the training loss of the three architectures. While the pure Transformer and Mamba models have a similar convergence, the hybrid Jamba (no MoE) has a lower loss throughout this run.

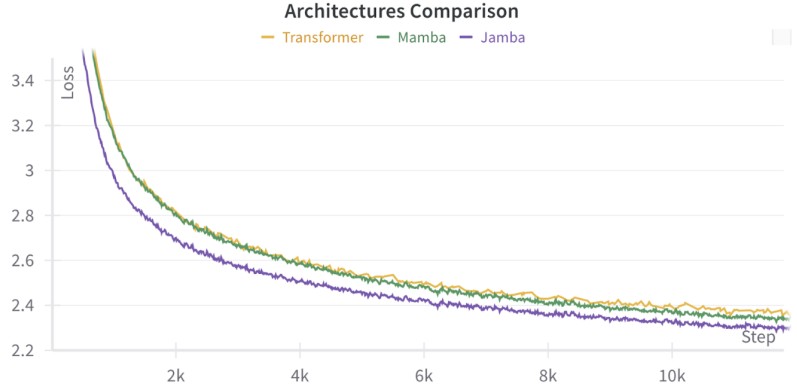

Figure 7: Training loss curves for pure Attention, pure Mamba, and an Attention-Mamba hybrid (no MoE). All models are 7B parameters. The hybrid achieves better loss throughout this training run.

## C.2 MAMBA-1 VS. MAMBA-2

We experimented also with Mamba-2 (Dao & Gu, 2024), a faster and improved version of Mamba, which was reported to outperform Mamba and Transformers separately. However, as Figure 8 shows, we found that in a hybrid architecture, the Mamba-1-Attention combination works better than Mamba-2-Attention. (We also found the hybrid architecture to outperform pure Mamba-2.) We hypothesize this is because some of the advantages of Mamba-2 over Mamba-1, in particular the ability to use a much larger state size, are less significant when we have full attention layers interleaved between the Mamba layers, as they can pool information from the entire context.

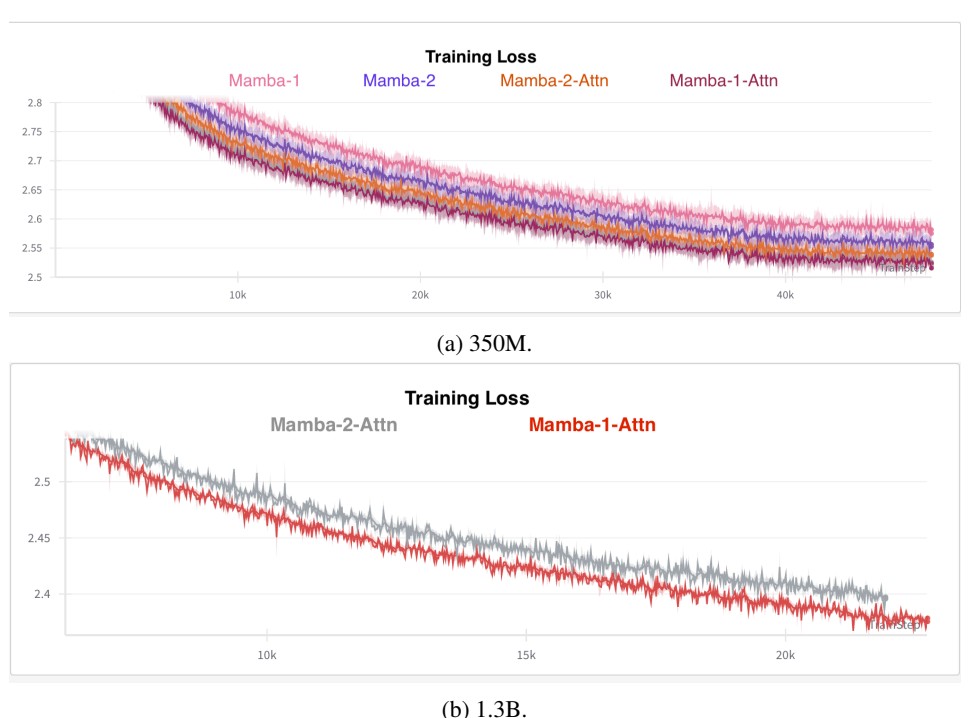

(a) 350M.

(b) 1.3B.

Figure 8: Comparison of Mamba-1, Mamba-2, Mamba-1-Attention, and Mamba-2-Attention on models trained on 100B tokens. While Mamba-2 outperforms Mamba-1 without attention, the hybrid Mamba-1-Attention performs better.

## C.3 WHY DOES THE COMBINATION WORK?

The pure Mamba model showed fairly good results in most tasks early on, including in general perplexity evaluations. However, it performed substantially worse than the pure Attention model in three common benchmark tasks: IMDB (Maas et al., 2011), QuAC (Choi et al., 2018), and NarrativeQA (Kocisky et al., 2018). In contrast, the hybrid Attention-Mamba performed similarly to the Attention model on these datasets. Table 9 shows the results for 1.3B models after 250B tokens.

Table 9: Mamba performs poorly on certain datasets, while the Attention-Mamba hybrid performs on par with the Attention model.

|                  | IMDB | QuAC | NarrativeQA |
|------------------|------|------|-------------|
| Attention        | 84.1 | 27.9 | 45.8        |
| Mamba            | 48.8 | 20.2 | 27.7        |
| Attention-Mamba  | 90.9 | 26.6 | 43.7        |

Looking into these results further, we found out that the pure Mamba model often does not follow the correct format. For instance, in the IMDB dataset, answer choices are "Positive" or "Negative". While the Attention model adheres to this format, the pure Mamba model often produces other answers, such as "Very Good", "Very Positive", "Funny", "Bad", "Poor", and "3/10". While these may be considered correct answers, the difficulty of Mamba to adhere to the format suggests a potential problem. Indeed, to perform successful in-context learning, it is important for models to capture the input-output format (Min et al., 2022). The hybrid Attention-Mamba model follows the format successfully, just like the pure Attention model.

We hypothesize that this phenomenon points to a limitation of SSMs – a potential difficulty in in-context learning (ICL). Indeed, the ability to perform ICL has been linked to the emergence of so-called induction heads in Transformer language models during training, which perform approximate copying operations that are supportive of ICL (Olsson et al., 2022). We conjecture that the lack of an attention mechanism in the pure Mamba model makes it difficult for it to learn in-context. While Mamba may learn to copy and perform simple ICL when explicitly trained to do so (Gu & Dao, 2023; Park et al., 2024), it is not clear if ICL is an emergent capability in SSM as is typical of Transformer models. In contrast, the hybrid Attention–Mamba model does perform successful ICL, even when only 1 out of 8 layers is an Attention one.

As anecdotal evidence of an emergent induction mechanism, we visualize in Figure 9 the attention of an example head from a 1.3B Attention-Mamba hybrid model (no MoE), on an IMDB example where the pure Mamba failed and the hybrid succeeded. Clearly, the attention from the last token (":") is focused on the labels from the few-shot examples. We have found 12 such heads in our hybrid model, in all three attention layers (which correspond to layers 4, 12, 20 in the model).

Future work can further investigate the emergence of ICL in hybrid models at large scale. Finally, recent work has attempted to extract attention-like scores from state-space models like Mamba (Ali et al., 2024), which opens another direction to search for induction capabilities in state-space models.

## C.4 THE EFFECT OF MIXTURE-OF-EXPERTS (MOE)

Recent work has shown that MoE improves Transformer language models while keeping compute manageable (Jiang et al., 2024).[*] However, it is not clear if MoE integrates well with state-space models at a large scale, and specifically with our hybrid Attention-Mamba architecture. Indeed, Table 10 shows that MoE improves the performance of the hybrid Attention-Mamba architecture at large scale (7B parameters trained on 50B tokens). The MoE variant has $n = 16$ total experts, $K = 2$ experts used at each token, and MoE is applied every $e = 2$ layers, as described in Section 3.

In terms of throughput/latency, the effects of MoE are linear with sequence length, as expected. Interestingly, compared to regular MLP layers, MoE has only about a 1.3–1.6x effect on throughput/latency, even though two experts are used (meaning 2x active parameters). This indicates that

---

[*]There is also initial evidence that MoE helps Mamba layers, albeit at small model and data scale (Pióro et al., 2024).

```
|BOS| _Passage : _To _call _this _film _a _disaster _will _be _an _understatement . _I _don ' t
_even _know _where _to _begin ! _I _have _questions _though , _and _lots _of _them . _I _would
```

**[...]**

```
_to _mention _a _few _of _course . _This _film _was _a _painful _experience _for _me _and _I
_advise _everyone _to _skip _it _by _all _means _necessary _and _possible . _Bollywood _should
_be _terribly _ashamed _of _this _kind _of _film – making . <|newline|> _Sent iment : _Negative
<|newline|> <|newline|> _Passage : _I _woke _up _and _it _was _a _beautiful _day ; _the _sun
_was _shining , _the _birds _were _singing _and _i _fanc ied _getting _a _movie , _something
```

**[...]**

```
_if _nothing _else , _OK _it _is _nothing _else . _Enjoy , _but _a _little _advice _– _before
_pressing _the _play _button _on _your _DVD _player , _throw _it _out _of _the _window .
<|newline|> _Sent iment : _Negative <|newline|> <|newline|> _Passage : _This _is _an _excellent
_James _Bond _movie . _Although _it _is _not _part _of _the _original _and _more _famous _series
, _and _it _is _a _standalone _film , _it _is _very _well _done . _En tic ing _Sean _Conner y
```

**[...]**

```
_different _version _of _Thunder ball , _updated _with _newer _technology . _Regardless _of _the
_repeated _theme , _there _are _sufficient _differences _to _make _it _most _entertaining . _I
_will _watch _this _one _frequently . <|newline|> _Sent iment : _Positive <|newline|>
<|newline|> _Passage : _I _am _a _fan _of _Ed _Harris ' _work _and _I _really _had _high
_expectations _about _this _film . _Having _so _good _actors _as _Harris _and _Von _Sy d ow _is
```

**[...]**

```
_movies _for _the _VHS _era _of _the _ 90 ' s . _Whatever _the _reason _was , _though , _this
_movie _was _a _very _bad _choice _for _anyone _involved . <|newline|> _Sent iment :
```

Figure 9: Example induction head (H3, first attention layer) from a hybrid Attention-Mamba model. Highlighted words reflect strong attention from the last token, ":", just before the model is about to predict the label. We see that the attention is focused on label tokens from the few-shot examples.

MLP/MoE layers are I/O bounded and since two experts can be computed independently, batching and fusing save time. In general, increasing MLP compute via MoE helps reduce the relative effect of Attention layers, which are quadratic in sequence length. In Jamba-1.5-Mini, Attention layers become dominant around a context length of 500K tokens, until then MoE is dominant. In Jamba-1.5-Large, where the matrices are larger, I/O is less of a problem, and MoE is the dominant factor. See Appendix C.8 for more results.

Table 10: Mixture-of-experts improves the Attention-Mamba hybrid.

| | OLLM | Hella Swag | Wino Grande | NQ | BoolQ | NQA | ARC-C | IMDB | log-prob C4 | Books | Code |
|---|---|---|---|---|---|---|---|---|---|---|---|
| w/o MoE | 36.6 | 62.5 | 58.8 | 15.4 | 59.4 | 38.3 | 35.0 | 87.1 | -0.547 | -0.658 | -0.340 |
| w/ MoE | 38.1 | 66.0 | 61.2 | 18.9 | 64.9 | 40.0 | 38.5 | 89.6 | -0.534 | -0.645 | -0.326 |

## C.5 STABILIZING MAMBA AT LARGE SCALE

When training Jamba models of up to 1.3B parameters, we observed stable training without special problems. However, when scaling to the largest model released here (7B-based, which has 12B/52B active/total parameters), we encountered large loss spikes. Investigating this revealed that inner parts of the Mamba layers suffer from large activation values, leading to the spikes. We therefore added RMSNorm (Zhang & Sennrich, 2019) to internal activations. As Figure 10 shows, this stabilized training and prevented additional loss spikes.

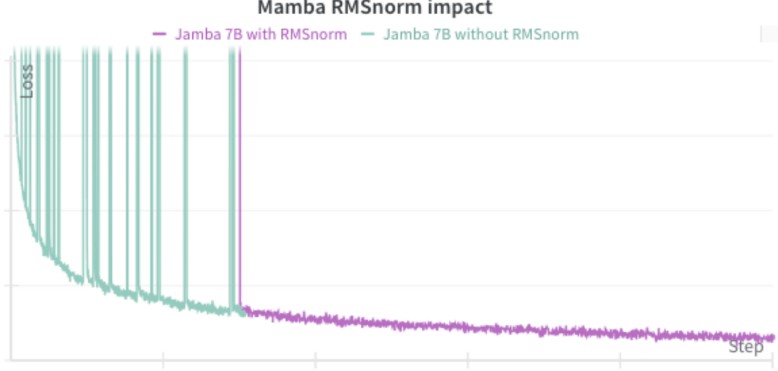

Figure 10: Adding RMSNorm to Mamba layers prevents loss spikes.

## C.6    JAMBA DOES NOT REQUIRE EXPLICIT POSITIONAL INFORMATION

Table 11 shows results of the Jamba architecture (with MoE) with no positional information and when applying RoPE (Su et al., 2024) in the attention layers (1.3B parameter models, 250B tokens). The results are similar, suggesting that explicit positional information may not be required for the hybrid architecture. Presumably, the Mamba layers, which are placed before attention layers, provide implicit position information.[*]

Table 11: Comparison of Jamba with and without explicit positional information.

|  | OLLM | Hella Swag | Wino Grande | NQ | BoolQ | NQA | ARC-C | IMDB | log-prob C4 | Books | Code |
|---|---|---|---|---|---|---|---|---|---|---|---|
| w/o RoPE | 39.6 | 71.5 | 64.2 | 22.2 | 68.9 | 50.5 | 40.7 | 93.1 | -0.516 | -0.623 | -0.299 |
| w/ RoPE | 40.1 | 71.8 | 65.5 | 22.2 | 67.9 | 46.2 | 40.4 | 90.7 | -0.516 | -0.623 | -0.299 |

## C.7    ADDITIONAL QUANTIZATION RESULTS

Table 12 provides results (with Jamba-1.5-Mini) with ExpertsInt8 compared to no quantization, showing that our quantization technique does not hurt quality. Figure 11 shows throughput measurements to complement the latency measurements in Section 4.1.

Table 12: Comparison of Jamba-1.5-Mini with and without ExpertsInt8 on academic benchmarks, showing no noticeable performance degradation.

|  | GSM8K | MMLU | NarrativeQA | log-prob (C4) |
|---|---|---|---|---|
| w/o ExpertsInt8 | 59.5 | 67.3 | 67.8 | -0.4301 |
| w/ ExpertsInt8 | 59.8 | 67.3 | 68.7 | -0.4301 |

## C.8    ARCHITECTURE EFFECTS ON LATENCY AND THROUGHPUT

In this section we provide ablations of various aspects of the Jamba architecture in terms of latency and memory uasge. We compared multiple variants to the Jamba architecture reported in the main paper along three metrics: encoding time, decoding time, and number of active parameters. In each variant, we changed one of the following: ratio of Attention-to-Mamba layers $a : m$, applying MoE every $e$ layers, total number of experts per layer $n$, number of top experts to use $K$, number of layers in a Jamba block $l$, and using MoE vs. dense layers.

---

[*]Some prior evidence suggested that Transformer decoder models do not need positional encodings (Haviv et al., 2022). However, all existing large-scale models do use some sort of explicit position information.

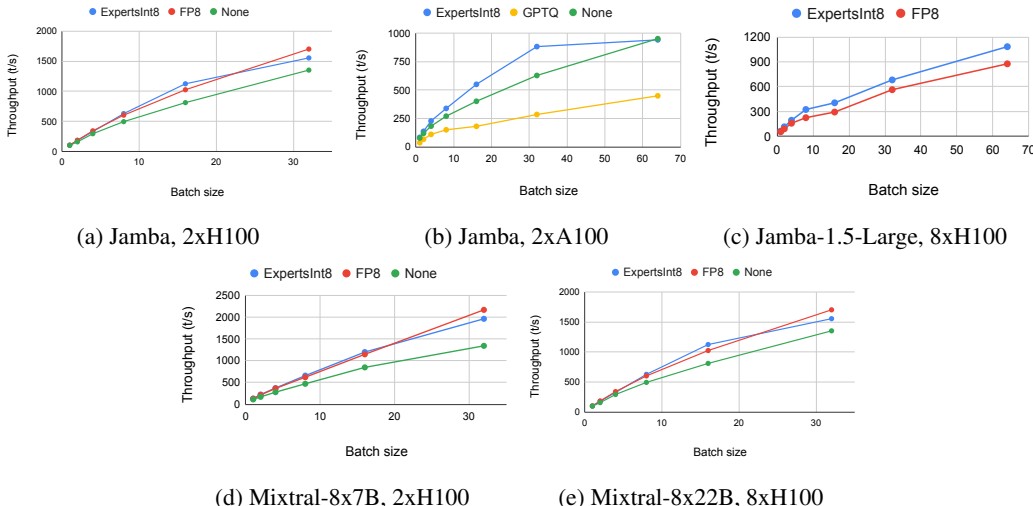

(a) Jamba, 2xH100    (b) Jamba, 2xA100    (c) Jamba-1.5-Large, 8xH100

(d) Mixtral-8x7B, 2xH100    (e) Mixtral-8x22B, 8xH100

Figure 11: Comparison of different quantization techniques. Output throughput with 1024-token context and 128-token decoding.

The main conclusions from these ablations are:

- Encoding latency seems consistent across different context lengths.
- Decoding times are flat up to about 64K-token context, starting to vary after that.
- Dropping MoE leads to gains at decoding time in longer contexts, up to 30% at 128K-token context.
- Using MoE at every layer instead of every two layers increases the active parameter count by 23% while slowing encoding/decoding by 11/14%, respectively.
- Doubling the attention-to-Mamba ratio slows decoding by 50% with a context of a 64K tokens.
- Using 16 MoE experts at every other has similar memory requirements to 8 experts at every layer, with lower latency and similar quality.
- Each MoE layer adds about 2.7B parameters. Overall, memory is 5.5x higher with MoE.
- Using MoE at every layer instead of every other layer increases paramerter count from 52B to 95B in Jamba-1.5-Mini. Using 8 experts instead of 16 lowers this number to 29B. Using 8 experts at every layer (similar to Mixtrla-8x7B) slightly lowers memory requirement (49B compared to 52B parameters), but with higher latency.
- The attention-to-Mamba ratio $a : m$ has a small effect on parameter count; attention-only takes 49.8B instead of 51.6B parameters. However, attention obviously has a high memory requirement at inference, as discussed in detail in Sections 2 and 3.

## D  POST-TRAINING DETAILS

During training, we trained with the normal next-token prediction loss, taken only on the assistant's tokens.

### D.1  SYNTHETIC DATA GENERATION

Section 6.3 described the general processes for post-training, including the use of synthetic data. In general, for synthetic completions when we know the answer, we use programmatic tools to verbalize it into a textual answer. In more complex cases, such as when generating a document, we use a mix of early in-house versions of Jamba and open models with permissive weights. We filter generated completions using both programmatic metrics and reward models, which allows for using smaller models by repeated sampling and filtering.

Here we give several notable examples of synthetic data generation:

**Table-based QA.** We generate tabular data and accompanying question-answer pairs, as demonstrated in our work on table understanding (Lenz et al., 2024). We then convert the tables into natural language paragraphs using a language model. Our generated training examples include extraction, aggregation, and attribution tasks vis-a-vis text corresponding to specific rows or columns in a given table.

**Document QA.** Given a document, we prompt a language model to generate question-answer pairs, for both single and multiple paragraphs. We sometimes embed these examples within longer context by adding similar texts, to encourage long-context understanding with attribution.

**Tool use.** We use the open-source Glaive function-calling dataset (AI) as a starting point, filtered with various heuristics and validations on the output schemas. To support parallel function calling, we first generate multiple valid parameter assignments for each function in Glaive. Next, we sample subsets of these valid parameter assignments, for the same function and across different functions, to generate user requests corresponding to the set of function calls. Finally, we prompt a function-calling language model to respond to these generated user requests and retain only responses where the function calls matched the original parameter assignments.

**Steerability.** We defined a set of instructions that can be easily validated and synthesized prompts that include a generic document-drafting task with one or more constraints added to it. We generated completions for these prompts from a language model and used rejection sampling based on the validations of our fine-grained instructions plus a general-purpose reward model. To support instructions in system messages, we chose multiple prompts of this kind that share a fine-grained instruction instance and reformatted these prompts into a multi-turn conversation, with the instruction moved to the system message.

## D.2 SOME OBSERVATIONS

We share a few observations from the development of Jamba-1.5. While these are not fully explored, we hope they would inspire the community to look further into these issues.

First, while we included only a very small fraction of non-english data, for a few languages and only for specific skills in the post-training phase, our Jamba-1.5 models perform quite well in multiple languages. We did include multilingual data in the pre-training phase, as mentioned above. Thus we speculate that the models are able to use the learned knowledge from that phase when being post-trained mostly in English.

Second, our efficient Jamba architecture lowers the cost of fine-tuning on long contexts, allowing us to experiment more with a given budget. Thus we could experiment with multiple different training recipes at the post-training stage.

Finally, while preference tuning algorithms like PPO (Schulman et al., 2017) or DPO (Rafailov et al., 2024) improve alignment between model outputs and human intent, we found that the combination of careful synthetic data generation, data filtering, and supervised fine-tuning is crucial for obtaining a strong post-trained model.

## E ADDITIONAL RESULTS

### E.1 FULL RULER RESULTS

Table 13 provides the full results on the RULER benchmark, to complement the results in Section 7.1.

Table 13: Comparison of Jamba-1.5 models with other publicly available and proprietary models on the RULER benchmark. Results for other models are from the RULER Github. Jamba-1.5 models are the only ones with a confirmed effective length of 256K tokens.

| | Claimed Length | Effective Length | 4k | 8k | 16k | 32k | 64k | 128k | 256k | Avg. |
|---|---|---|---|---|---|---|---|---|---|---|
| **Jamba-1.5-Large** | 256K | **256K** | 96.7 | 96.6 | 96.4 | 96.0 | 95.4 | 95.1 | 93.9 | 95.7 |
| **Jamba-1.5-Mini** | 256K | **256K** | 95.7 | 95.2 | 94.7 | 93.8 | 92.7 | 89.8 | 86.1 | 92.6 |
| Gemini-1.5-pro | 1M | >128K | 96.7 | 95.8 | 96 | 95.9 | 95.9 | 94.4 | - | 91.4 |
| GPT-4-1106-preview | 128K | 64K | 96.6 | 96.3 | 95.2 | 93.2 | 87 | 81.2 | - | 91.6 |
| Llama 3.1 70B | 128K | 64K | 96.5 | 95.8 | 95.4 | 94.8 | 88.4 | 66.6 | - | 89.6 |
| Qwen2 72B | 128K | 32K | 96.9 | 96.1 | 94.9 | 94.1 | 79.8 | 53.7 | - | 85.9 |
| Command-R+ | 128K | 32K | 95.6 | 95.2 | 94.2 | 92 | 84.3 | 63.1 | - | 87.4 |
| Llama 3.1 8B | 128K | 32K | 95.5 | 93.8 | 91.6 | 87.4 | 84.7 | 77 | - | 88.3 |
| Command-R | 128K | 32K | 93.8 | 93.3 | 92.4 | 89.5 | 84.9 | 76 | - | 88.3 |
| Mistral Large 2 | 128K | 32K | 96.2 | 96.1 | 95.1 | 93 | 78.8 | 23.7 | - | 80.5 |
| Mixtral 8x22B | 64K | 32K | 95.6 | 94.9 | 93.4 | 90.9 | 84.7 | 31.7 | - | 81.9 |
| Yi 34B | 200K | 32K | 93.3 | 92.2 | 91.3 | 87.5 | 83.2 | 77.3 | - | 87.5 |
| Phi3 mini 3.8B | 128K | 32K | 92.2 | 91.5 | 90.7 | 87.5 | 80.6 | 66.7 | - | 84.8 |
| Phi3 medium 14B | 128K | 32K | 93.3 | 93.2 | 91.1 | 86.8 | 78.6 | 46.1 | - | 81.5 |
| Mixtral 8x7B | 32K | 32K | 94.9 | 92.1 | 92.5 | 85.9 | 72.4 | 44.5 | - | 80.4 |
| Mistral Nemo 12B | 128K | 16K | 87.8 | 87.2 | 87.7 | 69 | 46.8 | 19 | - | 66.2 |
| DBRX | 32K | 8K | 95.1 | 93.8 | 83.6 | 63.1 | 2.4 | 0 | - | 56.3 |

