# OpenReview forum: "Jamba: Hybrid Transformer-Mamba Language Models"
_ICLR.cc/2025/Conference — ICLR 2025 Poster_

### Official Review · Reviewer_pW77 · 2024-10-29

**Soundness:** 2
**Presentation:** 2
**Contribution:** 3
**Rating:** 6
**Confidence:** 4

**Summary:**

The paper introduces a model family of Transformer-Mamba models, augmented with Mixture of Expert (MoE) layers. Due to the Mamba and MoE components, the model has throughput and memory benefits on long context tasks. The authors describe a quantization method for the MoE layers which further enhances the inference performance. Finally, the authors describe some evaluations for long context, chat, and natural language benchmarks.

**Strengths:**

- The paper proposes a large-scale Hybrid of Mamba, Transformers, and MoE layers. The large-scale aspect of it with respect to model size is certainly novel.
- Although quantizing the Expert layers on MoE models has been explored before [1] [2] the quantization on Int8 is novel and interesting.

[1] https://neurips2023-enlsp.github.io/papers/paper_81.pdf

[2] https://arxiv.org/abs/2406.08155

**Weaknesses:**

Unfortunately there are many key aspects missing from all the sections of the paper which makes it hard to quantify if the proposed approach is better than the state of the art in an apples to apples setup, or for future reproductions and/or comparisons. The most important weakness is the lack of disclosing the pre-training size (or FLOPs budget in general), which is known to be indicative of scaling trends and allows for comparing models in an standardised way [5]. While I believe the proposed protocol is novel and interesting enough, I think the following weaknesses should be addressed so that this paper is useful for the ICLR community.

**Efficient Inference details**

- Section 4.1 and 5 focus on the latency aspects of the proposed quantization method, but there is no empirical evidence to quantify the loss in model performance (if any).
- Figure 2 from section 4.1 seems focused on latency and missing throughput, which is an important consideration to compare to the other approaches.
- Relevant very similar work in MoE quantization on the experts [2] is missing from section 4.1, there should be at least a discussion, at best a latency comparison.
- Section 5 lacks important details of what software stack was used to run the latency and throughput comparisons. It is not possible to assess to what degree the comparison is fair, or if it paints a good or bad picture for the proposed method. Concretely: what quantization is used in the baseline models if at all? What’s the software stack used for running these comparisons?

**Pre-training details**

- There are no meaningful details about the composition of the pre-training dataset or the ablations conducted, which makes it hard for future work to compare to this model (e.g. was Jamba trained strongly on X or Y language, or domain).
- There is no indication on section 6 or the rest of the paper about either the pre-training dataset size, or the amount of tokens the model saw during pre-training. Without this information, it is simply not possible to understand if the proposed model outperforms in a FLOPs apples-to-apples comparison with other models. This makes it hard for future work to make a call of whether or not it is worth it to implement the Jamba architecture with a given FLOPs budget — which is a tradeoff often times researchers in Academia and Industry must think about.
- MoE models are known to be finicky to pre-train [3], there’s no details about the MoE pre-training protocol. Such as for example, routing loss function and/or hyperparameters. Broadly speaking, there’s no discussion about pre-training hyperparameters at all, which makes it hard to reproduce Jamba in the future, or to run sensible comparable ablations.
- There’s no information about the tokenizer training either: the mixture of corpus size used, or software stack which is known to matter in practice [4].
- There is no mention at all about the pre-training effective throughput and FLOPs per step. This makes it hard for future work and practitioners to decide whether the proposed approach scales once the model performance is viewed as a function of FLOPs and wall-clock time.
- [1] is a very close recent work but is not discussed or ablated throughout the paper.

**Post-training details**

- There are no details at all about what algorithm was used during Post-training and what are their hyper-parameters and compute budget, which makes it hard to reproduce and to compare to other models.
- There’s a small mention of synthetic data, but there are no details about what model was used to bootstrap the synthetic data, the size, and under what protocol. This makes it hard to assess to what degree this model can be a distillation of other models.

**Evaluation**

- Section 7 has this statement “We mainly compare with recent open-weight models of the same size range” . This is problematic because the delta in performance is dictated by both model size **and** data budget [5]. Since the full compute budget is not disclosed, it’s hard to draw conclusions that can inform future research ideas and questions (Eg is the Jamba architecture better than transformers for a given compute budget).
- There are no details about the inference used to compute the self-reported numbers in table 4. This is important since some models/tasks are sensible to the use of different floating point precisions and quantization schemes. This is also important to guarantee a fair comparison.
- A seemingly arbitrary set of (model, tasks) are self-reported, whereas others are drawn from previous papers (without citation) or from the Huggingface OpenLLM Leaderboard. It is not clear why this is the case, which makes it harder for future work to reference this table or to compare to it.
- Since some numbers are self-reported and for the rest there’s no citation of where they come from (except for the leaderboard) it is not clear at all if this is a fair comparison under the same setup: prompts, split sets, same shot number. While this is a very challenging area of the LLM literature, there should be at least an explanation of the setup used to evaluate Jamba so that future work is somewhat comparable.
- The claim on Figure 5 seems overreaching, since Jamba 1.5-large is both larger in total and active parameters than llama 3.1 70b, and is well above taking into account error bars, so is not either of a similar size (the other models are smaller) or having competitive performance.

**Typos**

- quantization technique that **allows**
- Section 7.1.1: The results for key **proprietary**

[1] https://arxiv.org/abs/2402.01771

[2] https://neurips2023-enlsp.github.io/papers/paper_81.pdf

[3] https://arxiv.org/abs/2202.08906

[4] https://aclanthology.org/2024.findings-naacl.247/

[5] https://proceedings.neurips.cc/paper_files/paper/2022/file/c1e2faff6f588870935f114ebe04a3e5-Paper-Conference.pdf

[6] https://arxiv.org/abs/2404.06654

[7] https://arxiv.org/abs/2403.05530

**Questions:**

- Is there a regression downstream performance when using ExpertsInt8 Quantization?
- Why adding Gemma on Table 3 if it does not support the context window for this task?
- Section 7.2 begins with “While not our main focus” — what’s the main focus? This seems important to contextualize to what degree this model is applicable for future work.
- What is the rationale to self-report some metrics and models in table 4?
- What does strict/flexible mean in table 4?
- Why are jamba models having higher error bars than the other models in figure 5?
- There's a couple of ablations in the appendix comparing Jamba to vanilla transformers (with and without MoE) but it lacks the details to assess if it was a fair comparison. For example, a transformer and mamba models are notably different architectures, are the optimization parameters tuned for each model in figure 6 and 7? (Learning rate, scheduler, beta values if Adam, etc). Also, are the two models parameter count equivalent? If so, how is this compensated?

---

> ### Author Response · Authors · 2024-11-19
> **Response to review (part 1)**
>
> Thank you for your detailed review. We’re glad you appreciated the “memory and throughput benefits on long context tasks”, the large-scale aspect of our work, and that you found the quantization “novel and interesting”.
>
> Concerning the main weakness:  **“many key aspects missing from all the sections of the paper”** and **“most important weakness is the lack of disclosing the pre-training size (or FLOPs budget in general)”**
>
> We agree that our inability to disclose full details about model training is a limitation of this work. We have added a Limitations section at the top of the appendix that acknowledges this. Nevertheless, we emphasize that the novel aspects of this work, namely the new Jamba architecture and its efficient performance at long-contexts, are independent of these questions. We show evidence for this by providing ablations (Appendix B) of different model configurations when pre-training on *exactly the same data with exactly the same compute and evaluation budget*.
> That said, following your question, we have added additional details (Section 6.2 and 6.3) about the pre-training and post-training procedures, including the major pre-training data types and the pre-training context length. We have also added details about the post-training process.
>
> We believe our results and analyses provide valuable insights for the ICLR community, which is why we have decided to go through the peer-review process, in contrast to other work at this scale, which does not bother to engage with the ICLR community.
>
> Below we address the points you raised one by one.
>
> # Efficient inference details
> **1. “no empirical evidence to quantify the loss in model performance”** of the proposed quantization method
>
> As mentioned in Section 4.1, our quantization technique incurs no loss in model quality. Following your question, we have added in Appendix C.7 empirical evidence supporting this statement.
>
> **2. “Figure 2 from section 4.1 seems focused on latency and missing throughput”**
>
> Throughput and latency are naturally inversely related. Following your question, we have run additional throughput measurements. As Appendix C.7 now reports, we observe similar trends in throughput, supporting the benefits of ExpertsInt8.
>
> **3. “Relevant very similar work in MoE quantization on the experts [2] is missing”**
>
> Thank you for this reference. MoQE is focused on very low-bit quantization, which incurs a loss in quality (they report BLEU on machine translation). The other work you cited (Li et al.), which is based on GPTQ, is also focused on very low-bit quantization and doesn’t compare with a non-quantized model. It’s also from June 2024, which was too recent for us to directly compare with. Importantly, we provide a public implementation of our quantization directly in vLLMs’ fused_moe kernel. We have added mention of these two papers in footnote 5.
>
> **4. “Section 5 lacks important details of what software stack was used to run the latency and throughput comparisons”** and **“what quantization was used in the baseline models if at all”**
>
> All cases were tested using the same stack of software (vLLM image == 0.5.5) , same hardware (H100/A100, depending on the chart), and same code for testing (vLLM latency test). We did not quantize any of the models, except for Jamba-Large (with ExpertsInt8) and LLaMA-3.1-405B (with FP8), since both do not fit on a single 8xA100 node without quantization. We managed running LLaMA-3.1-405B on A100 with FP8, but reached only a sequence length of ~100K tokens, hitting out-of-memory after that. In contrast, Jamba-Large can reach 256K tokens.

---

> > ### Comment · Reviewer_pW77 · 2024-11-21
> > **Appendix C.7**
> >
> > Thanks a lot for adding Table 12! Would you have the same numbers for the larger Jamba models? I've seen typically quantization methods to break down with big models and long context lenghts (NarrativeQA is interesting in this regard). Adding the VLLM image details in the manuscript would be useful too (or point to a repo where this info is centralized if that's more convenient)

---

> > > ### Author Response · Authors · 2024-11-24
> > >
> > > Unfortunately we don't have this number. Running Jamba-Large at inference without quantization is not trivial as it requires multi-node inference. We did have early results showing comparable quality with and without quantization. We will try to get such results soon.
> > >
> > > As for vLLM: we removed the link to the specific version to maintain anonymity. It will be made available after the reviewing is done.

---

> > > > ### Author Response · Authors · 2024-11-25
> > > > **Quantization results with Jamba-Large**
> > > >
> > > > We were able to run Jamba-Large without quantization with multi-node inference. Here are the results with and without quantization, showing no noticeable difference in quality:
> > > >
> > > > |                 | GSM8K | MMLU | NarrativeQA | log-prob (C4) |
> > > > | ------      | ---- | ----- | ------ | ------ |
> > > > w/o quant. |  78.4 |	79.4  | 80.3 | -0.393 |
> > > > w/ quant.   | 79.1  |	79.3 |  80.2 | -0.393 |
> > > >
> > > > Note that these are results with an earlier version of Jamba-Large, so we prefer not to include them in the revision. The point is that quantization does not hurt quality even at this scale.

---

> > > > > ### Comment · Reviewer_pW77 · 2024-11-26
> > > > > **Re: Quantization results with Jamba-Large**
> > > > >
> > > > > Since these results are already in the appendix, I suggest adding them with a footnote clarifying how you did the experiment (since you did, regardless of the model version, and it does seem to work, and I think we both can agree the model version shouldn't change the quantization results too much, the scores here suggest the model is already deep into training, regardless whether it's the final large model or not)

---

> ### Author Response · Authors · 2024-11-19
> **Response to review (part 2)**
>
> # Pre-training details
> **1+2. “no meaningful details about the composition of the pre-training dataset”** and **“pre-training dataset size, or the amount of tokens the model saw during pre-training”**
>
> As mentioned above, we acknowledge this limitation of our work, and have added some details following your question.
>
> However, we disagree with your statement that “This makes it hard for future work to make a call of whether or not it is worth it to implement the Jamba architecture with a given FLOPs budget”. As clearly shown in the ablations in Appendix C (previously Appendix B), training the Jamba architecture with a given FLOPs budget leads to strictly better results than various alternative configurations (vanilla Mamba or Transformer, no-MoE, other Attention-to-Mamba ratios). Thus, our ablations provide valuable evidence for the ICLR community in designing and implementing future hybrid models.
>
> **3. “no details about the MoE pre-training protocol.”**
>
> We follow standard protocols for MoE training, as mentioned in Appendix B (previously Appendix A). In particular, we used standard router load-balancing loss to balance the routing decisions and a router zloss to avoid large router logits. We’ve added this information to Appendix B. The number of total and active experts is specified in Section 3.
>
> **4. “No information about the tokenizer training”**
>
> We used a similar mixture corpus for training the tokenizer as in our pre-training mixture.
> We trained with the same software and very similar settings as the LLaMA/Mistral tokenizers, namely, SentencePiece BPE, which was shown [4] to be better than HuggingFace BPE. We’ve added this information to Appendix B (previously Appendix A). As noted in the Appendix, we set numbers to single digits as is common in recent models.
>
> **5. “pre-training effective throughput and FLOPs per step”**
>
> The important question to answer is how efficient the Mamba and Attention layers are during pre-training. We evaluated this with the models FLOPs utilization metric (MFU), proposed in the PaLM paper. In our measurements, the MFU with vanilla attention drops from 60% to 20% at 32K token length, while the Mamba MFU remains close to 60%. In longer contexts, the difference would be even bigger.
>
> **6. “[1] is a very close recent work”**
>
> Thank you for this reference to BlackMamba. We were not aware of this work and have added mention of this to Section 1. One small difference is that BlackMamba applied MoE to MLP, while we apply it only at every other layer. And, of course, BlackMamba is not a hybrid Attention-Mamba model.
>
> # Post-training details
> **1. “no details at all about what algorithm was used during Post-training and what are their hyper-parameters and compute budget”**
>
> As Section 6.3 and Appendix C.2 (previously Appendix B.2) indicate, our training algorithm was supervised fine-tuning. While we are not able to provide details on hyper-parameters and compute budget, we share interesting observations in Appendix C.2, which we hope will be useful for the community.
>
> **2. “no details about what model was used to bootstrap the synthetic data, the size, and under what protocol”**
>
> Appendix D (previously Appendix C) provides some information about the synthetic data generation protocol. We have now added additional details on our general strategy of sampling and filtering.

---

> > ### Comment · Reviewer_pW77 · 2024-11-21
> > **RE: Pretraining budget**
> >
> > My comment regarding the FLOPs budget it's also directed are general scaling trends. While a given architecture might work at a given FLOPs budget, it might not have a good performance as you scale up (or down). One final point it might be worth to make is that, assuming that is not possible to disclose how well the architecture scales as you go from 1b-7b/250b to 94B/?, it'd be interesting if you have results that go in the opposite direction: below 1b-7b parameters and/or 250b tokens, and draw a trend (I'm speculating you did smaller ablations). If that data is available it's valuable for future scaling laws research.
> >
> > While MFU is important, the throughput matters as well, specially since the amount of accelerators is not disclosed: you might have a relatively bad MFU but faster training just because you have a lot of devices available. If you have a fixed budget of days for a given cluster for a single run, you will want to maximize your throughput, and wouldn't use less devices solely because it maximizes MFU, or if you maximize MFU, you'd try to find a model size that fits better both constraints. On any case, it would be very valuable for systems research if you add the MFU numbers to the manuscript.

---

> > > ### Author Response · Authors · 2024-11-24
> > >
> > > Unfortunately we don't have detailed results for drawing scaling laws, although this is indeed very interesting. In general it is not straightforward to conduct such experiments reliably. One would need to decide on several training data sizes ahead of time (due to the learning rate schedule), and train multiple models at different sizes. We're looking into finding useful results from our existing measurements, and will update if possible. However, in general a detailed scaling laws study with the Jamba architecture would be a great topic for future work.
> > >
> > > We agree that MFU is not the only important factor because of the potential number of GPUs. While we don't have throughput numbers to provide, generally pre-training with Jamba is much more efficient than pure attention. In any case, we've added the MFU numbers to the new revision (Section B), as you requested.

---

> > ### Comment · Reviewer_pW77 · 2024-11-22
> > **Appendix D**
> >
> > Would it be possible to expand appendix D to describe which open-weight models were used? It's good for the field to properly credit when OSS models are used to contribute to new models, similar point for reward models (if open-weights models were used for this at all).

---

> > > ### Author Response · Authors · 2024-11-24
> > >
> > > I'm afraid we can't disclose this particular information about post-training.

---

> ### Author Response · Authors · 2024-11-19
> **Response to review (part 3)**
>
> # Evaluation
> **1. “hard to draw conclusions that can inform future research ideas and questions (Eg is the Jamba architecture better than transformers for a given compute budget).”**
>
> There are two kinds of evaluations and they serve different purposes. The evaluations reported in Section 7 are of a fully trained model. Here, we compare with other open-weight models, which are also similarly released as is. The purpose of these evaluations are to provide users and researchers with information when they choose which *model* to use as is or as a starting point.
>
> A separate kind of evaluation is to answer the question of which *architecture* to use. Here, indeed, it is important to compare under a given compute budget, which is precisely what the ablations in Appendix C (previously Appendix B) do. As explained above, these ablations support the benefits of Jamba over alternative configurations under a given compute budget.
>
> **2+3+4. “ no details about the inference used to compute the self-reported numbers in table 4.”** and **“A seemingly arbitrary set of (model, tasks) are self-reported, whereas others are drawn from previous papers (without citation) or from the Huggingface OpenLLM Leaderboard”** and **“not clear why this is the case”** and **“not clear at all if this is a fair comparison under the same setup”**
>
> Table 4 contains symbols indicating which evaluations were run by us and which were taken from the OpenLLM leaderboard. Results with competitors that are not marked are taken from the papers of these models. Whenever possible, we used previously reported numbers. When those were not available, we calculated them ourselves.
>
> We have made every effort to ensure the comparison is fair by using the same setup: prompts, official splits, and number of shots (which is indicated in the table). When evaluating Jamba and self-reporting results of other models, we always used the official repositories. We mainly used the lm-evaluation-harness (https://github.com/EleutherAI/lm-evaluation-harness) whenever possible. Regarding evaluations not in lm-evaluation-harness: For HumanEval, BFCL, and IFEval, we used the official dataset and metrics, as mentioned in their official repos. For RealToxicity, we used the official dataset and used their API for judgment. As you rightfully note, this is a very challenging area, but our comparison is as fair as possible.
>
> In terms of the inference for evaluation, we have used vLLM for all self-reported scores, no quantization for all models except for ExpertsInt8 quantization for Jamba-Large. If you have specific questions here, we would be happy to provide more information.
>
> **5. “The claim on Figure 5 seems overreaching”**
>
> We agree that LLaMA-3.1-70B is better than Jamba-Large, but note that Jamba-Mini is better than LLaMA-3.1-8B. We have revised the statement to make it more fair.
>
> **Typos**: fixed, thank you.

---

> > ### Comment · Reviewer_pW77 · 2024-11-21
> > **on evaluation metrics**
> >
> > Thanks a lot for clarifying the eval setup. I strongly suggest you mention and cite that you used the lm-evaluation-harness (they have a paper now too: https://arxiv.org/pdf/2405.14782), this is very important for reproducibility, and if possible specific commit revisions for future comparability. LLMs evaluation is very challenging but adding these small details go a long way. Same comment for the other codebases you mention that were used during evaluation.

---

> > > ### Author Response · Authors · 2024-11-24
> > > **Added to revision**
> > >
> > > Done. Added to Section B in the new revision.

---

> ### Author Response · Authors · 2024-11-19
> **Response to review (part 4)**
>
> # Questions
> **1. “Is there a regression downstream performance when using ExpertsInt8 Quantization?”**
>
> No. See new results in Appendix C.7.
>
> **2. “Why adding Gemma on Table 3 if it does not support the context window for this task?”**
>
> This was done just as a matter of style to conform with the other tables.
>
> **3. “Section 7.2 begins with “While not our main focus” — what’s the main focus? This seems important to contextualize to what degree this model is applicable for future work.”**
>
> The main focus is to provide long context capabilities with improved throughput, which is why Section 7.1 appears first. We provide standard academic benchmarks in Section 7.2 as a useful reference to compare with other models.
>
> **4. “What is the rationale to self-report some metrics and models in table 4?”**
>
> As explained above, we always used previously reported scores whenever they were available. We only self-report results that are missing from prior studies.
>
> **4. “What does strict/flexible mean in table 4?”**
>
> Strict and flexible are two common ways to evaluate on GSM8K. Since LLaMA-3.1 models perform poorly with the standard strict evaluation, we also reported for them the flexible metric, which allows for higher results. Strict means that the answer must follow “####” while flexible searches for the numbers in the completion. See the LM Evaluation Harness for the regexes: https://github.com/EleutherAI/lm-evaluation-harness/blob/main/lm_eval/tasks/gsm8k/gsm8k.yaml.
>
> **5. “Why are jamba models having higher error bars than the other models in figure 5?”**
>
> Jamba models are more recent than other models, so they have fewer votes on the ChatbotArena and thus higher error bars.
>
> **6. “There's a couple of ablations in the appendix comparing Jamba to vanilla transformers (with and without MoE) but it lacks the details to assess if it was a fair comparison.”**
>
> We have tuned each hyper-parameter separately for each architecture in an effort to get the best possible model. The models are also using a very similar parameter count. In general it’s not possible to use exactly the same parameter counts due to difference in architectures, but our models are very similar. Importantly, the boost in performance from pure Mamba to the hybrid Attention-Mamba comes with the hybrid having fewer parameters.
>
> We hope we were able to answer as many of your comments as possible, and that you now agree that the contributions of this work are valuable to the ICLR community.

---

> ### Comment · Reviewer_pW77 · 2024-11-26
> **Final comment**
>
> While I think it's unsatisfactory that the authors are not able to disclose what open models were used in Appendix D, general pre-training hyperparameters, and I share the concerns of reviews x4G4 (hope the AC weighs in), I think the authors have made a notable effort to improve the manuscript and  are genuinely engaged on sharing as much as they can -- an increasingly challenging point in industry research -- . Hence, I'm increasing my score. Thanks to the authors for engaging.

---

> > ### Author Response · Authors · 2024-11-26
> >
> > We're pleased we were able to mitigate some of your concerns and thank you for the discussion.

---

### Official Review · Reviewer_7oKN · 2024-10-30

**Soundness:** 3
**Presentation:** 3
**Contribution:** 2
**Rating:** 6
**Confidence:** 4

**Summary:**

This paper presents Jamba, a hybrid MoE model with interleaved Attention and Mamba layers. They open-source Jamba-1.5-Large (94B active parameters) and Jamba-1.5-Mini (12B active parameters). The authors conduct extensive experiments on various design choices, including different ways to interleave attention and Mamba layers and mix experts. Jamba demonstrates strong results and inference efficiency on long-context tasks, and is competitive on standard language tasks (e.g., MMLU). They introduce ExpertsInt8 to save time and memory of model loading for MoE models.

**Strengths:**

1)	The proposed model, Jamba, is open sourced for research use.
2)	Jamba achieves strong results on long-context tasks, while offering better inference efficiency on throughput and memory footprint.
3)	The authors provide detailed ablations and insights on different ways of combining attention and Mamba layers, including the ratio of attention and Mamba layers, using Mamba-1 or Mamba-2 layers.

**Weaknesses:**

Lack of a fair comparison on the effectiveness of Jamba against the other hybrid attention-SSM architectures. Although Jamba achieved promising results, the paper lacks a comparison with other hybrid attention-SSM models, such as YOCO [1] and Samba [2], using the same training data and parameter scale, particularly on the long-context and standard language tasks (e.g., MMLU).


[1] Sun, Yutao, et al. "You only cache once: Decoder-decoder architectures for language models."
[2] Ren, Liliang, et al. "Samba: Simple Hybrid State Space Models for Efficient Unlimited Context Language Modeling."

**Questions:**

Add more comparsion against the other hybrid attention-SSM models under a fair setting.

---

> ### Author Response · Authors · 2024-11-19
> **Response to review**
>
> Thank you for your helpful review. We’re glad you appreciated our “extensive experiments” with “strong results on long-context tasks” and “better inference efficiency”. We’re also happy you appreciate that Jamba is publicly available, and our “detailed ablations and insights”.
>
> We would like to address the weakness and question you mentioned.
>
> **“Lack of a fair comparison on the effectiveness of Jamba against the other hybrid attention-SSM architectures”** and **“comparsion against the other hybrid attention-SSM models under a fair setting”**:
>
> As discussed in the Introduction, all prior attempts with hybrid models are either of smaller scale or perform worse than vanilla Mamba or vanilla Transformers. Concerning YOCO and Samba, they were posted on arxiv in May and June 2024, which was too recent for us to compare with under the same setting. We have now added mention of these recent attempts to footnote 2 in the Introduction. We did try the main aspect that Samba notes, namely the sliding-window attention. In our preliminary experiments with a 1.3B-parameter model, combining Mamba with sliding window attention performed worse than Mamba with attention without sliding window. (Note that the YOCO and Samba models themselves appear weaker than Jamba (e.g., on MMLU), but that is not a fair comparison, so we are not stressing this point.)
>
> We hope we have addressed your concern. Given the positive note of your review, we would appreciate it if you can consider increasing your score. Please let us know if you have any other questions.

---

> ### Author Response · Authors · 2024-11-24
>
> Dear reviewer, we hope our response and new revision has addressed the points you have raised. Could you let us know if you have any other comments or questions? Assuming we have answered you concerns, we would appreciate it if you can consider revising your evaluation.

---

### Official Review · Reviewer_x4G4 · 2024-11-02

**Soundness:** 2
**Presentation:** 3
**Contribution:** 2
**Rating:** 5
**Confidence:** 4

**Summary:**

The paper introduces Jamba-1.5-{Large, Mini} models based on a novel hybrid of Transformer, Mamba, and MoE architectures. Combining the architectural design and the novel ExpertsInt8 quantization technique, the models are efficient in serving in terms of latency, throughput, and memory footprint. Evaluations on long-context benchmarks demonstrate that the model has strong performance in its effective 256K context length. Academic and chat benchmark evaluations indicate that Jamba-1.5 models perform similarly to SOTA public models.

**Strengths:**

1. Serving the proposed Jamba architecture is efficient in terms of low latency, high throughput, and reduced memory footprint, especially in longer context length.
2. The paper shares some insights found in the pre-training process, which can be beneficial to the community.
3. The performance of Jamba-1.5 models is close to other public models of similar activated parameters.

**Weaknesses:**

1. The details of model training are not revealed, including data source, data mixture ratio, number of tokens trained, context length, and post-training techniques. This makes the training process unclear and not reproducible.
2.  A direct comparison between Jamba, Llama, and Mistral is insufficient to demonstrate the effectiveness of the proposed Jamba architecture.
	- The training of Jamba, Llama, and Mistral models are different in data, model size, and computation, making it difficult to attribute the evaluation results. A comparison between architectures (like Jamba, Mamba, and Attention; with and without MoE) under the same pre/mid/post-training condition (like with the same data and same computation or number of tokens) can better reveal the effectiveness.
	- Although Appendix B provides some ablation studies in architecture, the training computation, model size, and reported evaluation datasets are rather limited, and evaluation after mid/post-training is not included. This is insufficient to demonstrate the architectural effectiveness at scale and in comprehensive downstream tasks.
3. It would be better to demonstrate the scaling ability of the Jamba architecture. For instance, the relationship of LM loss/downstream task performance wrt. training computation.

**Questions:**

- Is it possible to provide details on training procedures? (e.g., pre-training: data source, number of tokens, training throughput, training time, training context length, training batch size; post-training: strategies used and tokens trained)
- Is it possible to provide experimental results of comparison concerning the Jamba/Mamba/Attention/MoE architectures under the same training setting at scale (like more tokens and larger model size) and for more comprehensive tasks (like all tasks in Section 7)?

---

> ### Author Response · Authors · 2024-11-19
> **Response to review**
>
> Thank you for your helpful review. We’re glad you noted the work’s strengths: the model’s “strong performance in its effective 256K context length”, “serving of the proposed Jamba architecture [...] efficient”, and the “insights shared during the pre-training process”.
>
> We address the weaknesses and questions below.
>
> **1. “The details of model training are not revealed”** and the question on **“details on training procedures”**
>
> We agree that our inability to disclose full details about model training is a limitation of this work. We have added a Limitations section at the top of the appendix that acknowledges this. Nevertheless, we emphasize that the novel aspects of this work, namely the new Jamba architecture and its efficient performance at long-contexts, are independent of these questions. We show evidence for this by providing ablations (Appendix C, previously Appendix B) of different model configurations when pre-training on *exactly the same data* with *exactly the same compute and evaluation budget*.
>
> That said, following your question, we have added additional details (in Section 6.2 and 6.3) about the pre-training and post-training procedures, including the major data types in pre-training and the pre-training context length. We have also added details about the types of data used in post-training. Refer also to Appendix D (previously Appendix C), where we have provided information and insights about post-training strategies.
>
> **2. “A direct comparison between Jamba, Llama, and Mistral is insufficient to demonstrate the effectiveness of the proposed Jamba architecture.”** and the question about **“experimental results of comparison…”**
>
> We agree that comparing different *released models* is insufficient to make claims about a proposed *architecture*. For this reason, we provide the ablation studies in Appendix C (previously Appendix B), as you acknowledge. These studies clearly justify our design choices in terms of the advantage of hybrid over vanilla models, which attention-to-Mamba ratio to use, and the benefit of MoE. The takeaways are summarized at the top of Section 3.
>
> We emphasize that these experiments were conducted under *exactly the same data and same computation* (number of tokens) for each compared variant. Thus, the comparison is fair. Moreover, the ablations are conducted at a fairly large scale: 1.3B/7B parameter models trained for 250B/50B tokens. This is a large scale for ablation studies that already required substantial compute and expenses.
>
> Finally, it is impossible to perform architectural ablations at mid/post-training, since this would require full pre-pretraining of multiple configurations, which is generally unrealistic.
>
> Moreover, the ablations in Appendix C (previously Appendix B) already include results from 3 academic benchmarks, the OpenLLM leaderboard summary (which is an aggregate metric over 7 datasets), and log-probs with 3 data sources of different domains. These were chosen because we found them informative for ablations at a smaller scale than full pre-training, since not all benchmarks provide informative results at this scale. Nevertheless, given your question, we have added additional ablation results to Appendix C on more tasks. Specifically, we added results on BoolQ, NarrativeQA, ARC-Challenge, and IMDB. While some of them are unstable (which we expanded upon in Appendix C.3), the results are in line with our prior observations.
>
> **3. “demonstrate the scaling ability of the Jamba architecture”**
>
> Thank you for this suggestion. Conducting a full scaling study is a complicated procedure taking many resources, which is beyond our scope. However, we are looking into providing measurements at multiple scales and will get back to you ASAP.
>
>
> We hope our response has helped answer your concerns. If you have any additional questions, please let us know. We would appreciate it if you can revise your evaluation accordingly.

---

> > ### Comment · Reviewer_x4G4 · 2024-11-26
> >
> > Thanks to the authors for the detailed responses.
> >
> > I reviewed Appendix C of the latest version, which justifies architectural choices and demonstrates the effectiveness of Jamba architecture with fair comparison. I believe my concerns about the architectural effectiveness of Jamba were largely addressed.
> > (By the way, it would be better to include a detailed setup for the ablation studies in Appendix C. For instance, the model architectural setting, like hidden size, number of total layers, the choice of optimizer and learning rate schedule, etc. So that the results are better reproducible.)
> >
> > However, for a paper introducing a new model architecture, the lack of scaling experiments and the inability to disclose pre-training and post-training details for the two main models are significant drawbacks. The lack of this information makes it difficult for the reader to interpret the results in the main part of the paper. I would like to retain my current assessment.

---

> ### Author Response · Authors · 2024-11-24
>
> Dear reviewer, we hope our response and the new paper revision have addressed your concerns. Please let us know if you have any remaining questions. We would appreciate it if you can go over our response and revision, and consider revising your evaluation assuming we have answered your concerns.

---

### Official Review · Reviewer_bBMs · 2024-11-04

**Soundness:** 3
**Presentation:** 4
**Contribution:** 4
**Rating:** 8
**Confidence:** 4

**Summary:**

The paper introduces Jamba, a hybrid language model combining Transformer, Mamba, and Mixture-of-Experts (MoE) layers to enhance memory efficiency, throughput, and long-context handling, supporting up to 256K tokens. With ExpertsInt8 quantization, Jamba achieves scalable, cost-effective deployment on 8-GPU setups. Experimental results demonstrate competitive performance across benchmarks in language modeling, chatbot, and multilingual tasks, highlighting its adaptability to various hardware and resource constraints.

**Strengths:**

1.Hybrid Architecture with Enhanced Flexibility : Jamba’s combination of Transformer layers, Mamba layers, and Mixture-of-Experts (MoE) modules offers a unique balance of efficiency and performance, addressing the limitations of each component individually. This hybrid design allows for configurable trade-offs between memory usage, throughput, and model capacity, making Jamba adaptable to diverse hardware configurations.

2. Long-Context Capabilities: Jamba supports an effective context length of 256K tokens, one of the longest among open-weight models. This is particularly advantageous for long-context tasks, as demonstrated in benchmarks like RULER and ∞BENCH, where Jamba-1.5 models perform competitively, underscoring their suitability for tasks requiring extensive memory.

3. Cost-Effective Inference via ExpertsInt8 Quantization
The introduction of ExpertsInt8 quantization allows Jamba-1.5-Large to fit on hardware with 8 80GB GPUs without compromising quality, even for long-context processing. This quantization technique shows latency advantages on A100 GPUs, where FP8 isn’t available, thereby providing a cost-efficient alternative to traditional approaches.

4.Competitive Performance Across Benchmarks
Jamba models perform comparably to state-of-the-art models on standard language modeling, chatbot evaluations, and multilingual benchmarks. This includes maintaining high throughput and latency efficiency, particularly at large context lengths, further demonstrating the model’s real-world applicability.

**Weaknesses:**

1. Minimal Discussion of MoE and Mamba Layer Interaction : While the hybrid design leverages both MoE and Mamba layers, the paper provides limited analysis of how these components interact to optimize performance. An in-depth exploration of how each component contributes to throughput, especially in long-context scenarios, would clarify the architectural benefits and limitations.

2. Sparse Performance Analysis on Edge and CPU: Although Jamba is designed to balance memory and compute efficiency, its performance on edge devices and CPUs is not evaluated. Given the trend toward deploying models in low-resource environments, this data would offer practical insights into the model’s viability beyond high-end GPU setups.

**Questions:**

In Section 3.2, the authors mention that “Jamba leverages a balanced mix of Transformer, Mamba, and Mixture-of-Experts (MoE) layers to optimize both throughput and model capacity, particularly in long-context scenarios.” It would be helpful if the authors could provide further insights into how the specific ratio of these layers was determined. An ablation study examining the impact of varying the number or sequence of each layer type (Transformer, Mamba, MoE) on key performance metrics such as latency, memory usage, and accuracy would lend greater clarity to the efficacy of this particular configuration and substantiate its contribution to Jamba’s overall efficiency.

---

> ### Author Response · Authors · 2024-11-19
> **Response to review**
>
> Thank you for your helpful review. We’re glad you appreciated the benefits provided by our “Hybrid Architecture with Enhanced Flexibility”, which “offers a unique balance of efficiency and performance”. We’re happy you noted the strengths of this architecture in “Long-Context Capabilities” and the “Cost-Effective Inference via ExpertsInt8 Quantization”, as well as the “Competitive Performance Across Benchmarks”.
>
> Below we address the weaknesses noted.
>
> **1. “Minimal Discussion of MoE and Mamba Layer Interaction”**
>
> Table 10 in Appendix B.4 provides a comparison of MoE and no-MoE hybrid models in terms of quality. The table shows improvements with MoE across the board.
>
> Considering the effect of throughput, we have conducted measurements and have added a discussion in the new version of the paper (Appendix C.4). The conclusion is that the effect of MoE on throughput/latency is linear in sequence length, as expected. Interestingly, compared to regular MLP layers, MoE has only about a 1.3/1.6x effect on throughput/latency in Jamba-Mini/Large, even though 2 experts are used (meaning 2x active parameters). This indicates that MLP/MoE layers are I/O bounded and since 2 experts can be computed independently, batching and fusing save time. In general increasing MLP compute via MoE helps reduce the relative effect of Attention layers, which are quadratic in sequence length. In Jamba-Mini, Attention layers become dominant around context length of 540K tokens, until then MOE is dominant. Attention starts to have a greater effect on throughput/latency compared to Mamba at ~300K tokens. In Jamba-Large, when the matrices are larger, I/O is less of a problem, and MoE is the dominant factor.
>
> **2. “Sparse Performance Analysis on Edge and CPU”**
>
> Thank you for raising this point. We find the prospects of the Jamba architecture for edge and CPU devices exciting. While implementing this would face some challenges (especially with custom Mamba kernels), we think this is a great avenue for future work.
>
>
> Regarding your question about **“how the specific ratio of these layers was determined”** and **"An ablation study […] on key performance metrics such as latency, memory usage, and accuracy”**:
>
> * Appendix C (previously Appendix B) provides a detailed ablation study in terms of quality. Appendix C.1 compares hybrid Attention-Mamba models to vanilla Attention or Mamba, including different ratios of Attention-to-Mamba. Appendix C.4 compares hybrid models with and without MoE.
> * In terms of latency and memory, we conducted detailed measurements and have added a new Appendix C.8 that reports our findings. Thank you for this suggestion.
>
> Thank you once again for your review and please let us know if you have any other questions or comments.

---

### Author Response · Authors · 2024-11-26
**General response; clarifying the contributions**

We thank the reviewers for their reviews and for engaging in discussion. The main weakness, pointed out by two reviewers, is the missing details on pre/post-training, such as data mix and training budget. We appreciate this perspective and have clarified as many details as possible during the rebuttal. We’re pleased we were able to mitigate some of the concern as reflected in the discussion and revised evaluation by one of the reviewers. However, we believe that these details are mostly **orthogonal to the main contributions of the paper**. We wish to emphasize the key contributions of this work:

1. We introduce a **and novel hybrid Transformer-Mamba-MoE architecture and demonstrate via careful and extensive ablation experiments its benefits over vanilla Transformer or Mamba architectures**. These ablations are done at large data and model scale, which required substantial resources and incurred high costs. Importantly, **the ablations are done in a fair comparison** using exactly the same training data and the same computation resources. This is a feature of our work, as many papers that introduce novel architectures do not report extensive ablations as we do (some don’t report any ablations). Moreover, we introduce a novel quantization technique to support efficient serving of large Jamba models and demonstrate its superiority over other quantizations with no quality degradation. In addition, we describe a novel loss we introduce to support efficient inference of Jamba models at large scale. Finally, we report throughput and latency measurements, which show the benefit of the Jamba architecture with long contexts. All these investigations and innovations are a beneficial contribution to the academic community.
2. We provide **implementations of Jamba as open-weight models of two large sizes**, along with an extensive evaluation on common benchmarks. This evaluation demonstrates the excellent performance of the released models in long context benchmarks, with high-quality on mainstream short-context benchmarks. Thus the released models are a useful artifact for the academic community to use and study.

We find the **requirement to disclose details such as the training data mix overly restrictive**. Open-weight models (Mistral, LLaMA) are the de facto standard for open models in the academic literature. However, papers describing such leading models do not disclose such details, not to mention closed API models (GPT, Claude). Indeed, there are numerous ICLR papers using LLaMA, Mistral, and similar models, although none of them were subject to the same requirements that our work is facing. Even the GPT3 paper, which won best paper at NeurIPS, did not disclose such information. At present there are only very few truly open-source models that disclose such information (e.g., OLMo) and they lag behind Jamba and other open-weight models, and are much less frequently used for academic work than, e.g., LLaMa or Mistral.

Moreover, most companies releasing frontier open-weight models **do not even bother to engage with the academic community and submit to peer review**, and instead only post technical reports on their websites. In contrast, we believe it’s important for papers about novel architectures and frontier open models to go through the peer-review process and engage with the academic discourse.

Penalizing this paper on the grounds of not disclosing the training information is not only orthogonal to the paper’s main contributions, but in fact means that **ICLR does not accept work describing novel architectures implemented as frontier models at large scale**. Subsequently, ICLR may be restricted to papers using existing released models, rather than introducing new ones.

In summary, we therefore contend that the position reflected in two of the reviews greatly misses the role of the ICLR community as we understand it.

---

### Meta-Review · Area_Chair_tdj5 · 2024-12-22

**Metareview:**

> The paper introduces Jamba, a hybrid language model combining Transformer, Mamba, and Mixture-of-Experts (MoE) layers to enhance memory efficiency, throughput, and long-context handling, supporting up to 256K tokens. With ExpertsInt8 quantization, Jamba achieves scalable, cost-effective deployment on 8-GPU setups. Experimental results demonstrate competitive performance across benchmarks in language modeling, chatbot, and multilingual tasks, highlighting its adaptability to various hardware and resource constraints.

Overall the results are good, but the experimental details are scarce and the technical decisions are not all backed by data. *I lean accept due to the performance but I am not sure it matches the scholarship required for ICLR*. Sadly this is a problem with the domain more than this single paper.

Reviewer x4G4 notes:
> The training of Jamba, Llama, and Mistral models are different in data, model size, and computation, making it difficult to attribute the evaluation results. A comparison between architectures (like Jamba, Mamba, and Attention; with and without MoE) under the same pre/mid/post-training condition (like with the same data and same computation or number of tokens) can better reveal the effectiveness.

And while this is true, a strict comparison is not really feasible in practice.

Reviewers x4G4 also discusses (in the rebuttal period):

> However, for a paper introducing a new model architecture, the lack of scaling experiments and the inability to disclose pre-training and post-training details for the two main models are significant drawbacks.

And the lack of scaling experiments is potentially the main drawback to propose a new architecture and convince the community.

**Additional Comments On Reviewer Discussion:**

Except for reviewer x4G4, there no discussion on rebuttal.

---

### Decision · Program_Chairs · 2025-01-22

Accept (Poster)